



**Analogue experiments on releasing and restraining bends and their application to**
**the study of the Barents Shear Margin**

**Roy H. Gabrielsen [1], Panagiotis A. Giennenas[2], Dimitrios Sokoutis [1,3], Ernst**
**Willingshofer[3], Muhammad Hassaan[1,4] & Jan Inge Faleide[1]**
[1] Department of Geosciences, University of Oslo, Norway
[2] Univ Rennes, CNRS, Geosciences Rennes, UMR 6118, 35000, Rennes France
[3] Faculty of Geosciences, Utrecht University, the Netherlands
[4] Vår Energi AS, Grundingen 3, 0250 Oslo, Norway
Corresponding author: Roy H. Gabrielsen (r.h.gabrielsen@geo.uio.no)
**ORCID iD:**
Jan Inge Faleide: 0000-0001-8032-2015
Roy H. Gabrielsen: 0000-0001-5427-8404
Muhammad Hassaan: 0000-0001-6004-8557



**Abstract:**

The Barents Shear Margin separates the Svalbard and Barents Sea from the North Atlantic. It includes one northern (Hornsund Fault Zone) and a southern (Senja Fracture Zone) margin segment in which structuring was dominated by dextral shear. These segments are separated by the Vestbakken Volcanic Province that rests in a releasing bend position between the two. During the break-up of the North Atlantic the plate tectonic configuration was characterized by sequential dextral shear, extension, contraction and inversion. This generated a complex zone of deformation that contain several structural families of over-lapping and reactivated structures Although the convolute structural pattern associated with the Barents Shear Margin has been noted, it has not yet been explained in this framework.

A series of crustal-scale analogue experiments, utilizing a scaled stratified sand-silicon polymer sequence, serve to study the structural evolution of the shear margin in response to shear deformation along a pre-defined boundary representing the geometry of the Barents Shear Margin and variations in kinematic boundary conditions of subsequent deformation events, i.e. direction of extension and inversion.

The observations that are of particular significance for interpretating the structural configuration of the Barents Shear Margin are:

1)The experiments reproduced the geometry and positions of the major basins and relations between structural elements (fault and fold systems) as observed along and adjacent to the Barents Shear Margin. This supports the present structural model for the shear margin.

2) Several of the structural features that were initiated during the early (dextral shear) stage became overprinted and obliterated in the subsequent stages.

3) Prominent early-stage positive structural elements (e.g. folds, push-ups) interacted with younger (e.g. inversion) structures and contributed to a complex final structural pattern.

4) All master faults, pull-part basins and extensional shear duplexes initiated during the shear stage quickly became linked in the extension stage, generating a connected basin system along the entire shear margin at the stage of maximum extension.

5) The fold pattern generated during the terminal stage (contraction/inversion became dominant in the basinal areas and was characterized by fold axes with traces striking parallel to the basin margins. These folds, however, most strongly affected the shallow intra-basinal layers.

This is in general agreement with observations in previous and new reflection seismic data from the Barents Shear Margin.





**Plain language summary:**

The Barents Shear Margin defines the border between the relatively shallow Barents Sea that is situated on a continental plate, and the deep ocean. This margin evolution history was probably influenced by the plate tectonic reorganizations. From scaled experiments we deducted several types of structures (faults, folds and sedimentary basins) that helps us to improve the understanding of the history of the opening of the North Atlantic.

**Key words:** Analogue experiments, dextral strike-slip, releasing and restraining bends, multiple folding, Barents Shear Margin**,** basin inversion





**Introduction**

Physiography, width and structural style of the Norwegian continental margin vary considerably along its strike (e.g. Faleide et al. 2008, 2015). The margin includes a southern rifted segment between 60° and 70°N and a northern sheared-rifted segment between 70° and 82°N (**Figure 1A**). The latter coincides with the ocean-ward border of the western Barents Sea and Svalbard margins (e.g. Faleide et al. 2008) and is referred to here as "the Barents Shear Margin". This segment coincides with the continent-ocean transition (COT) of the northernmost part of the North Atlantic Ocean, and its configuration is typical for that of transform margins where the structural pattern became established in an early stage of shear, later to develop into an active continent-ocean passive margin (Mascle & Blarez 1987; Lorenzo 1997; Seiler et al. 2010; Basile 2015; Nemcok et al. 2016).

Late Cretaceous - Palaeocene shear, rifting, breakup and incipient spreading in the North Atlantic was associated with voluminous magmatic activity, resulting in the development of the North Atlantic Volcanic Province (Saunders et al. 1997; Ganerød et al. 2010; Horni 2017). According to its tectonic development, the Barents Shear Margin (**Figure 1B**) incorporates, or is bordered by, several distinct structural elements, some of which are associated with volcanism and halokinesis.

The multistage development combined with a complex geometry caused interference between structures (and sediment systems) in different stages of the margin development. Such relations are not always obvious, but interpretation can be supported by the help of scale-models. In combining the interpretation of reflection seismic data and analogue modeling, therefore, we investigate structures generated in (initial) dextral shear, the development into seafloor spreading and subsequent contraction in this process, the later stages of which were likely influenced by plate reorganization (Talwani & Eldholm 1977, Gaina et al. 2009, see also see also Vågnes et al. 1998; Pascal & Gabrielsen 2001; Pascal et al. 2005; Gac et al. 2016) or other far-field stresses (Doré & Lundin 1996; Lundin & Doré 1997; Doré et al. 1999; 2016; Lundin et al. 2013). The present experiments were designed to illuminate the structural complexity affiliated with multistage sheared passive margins, so that the significance of structural elements like fault and fold systems observed along the Barents Shear Margin could be set into a dynamic context.



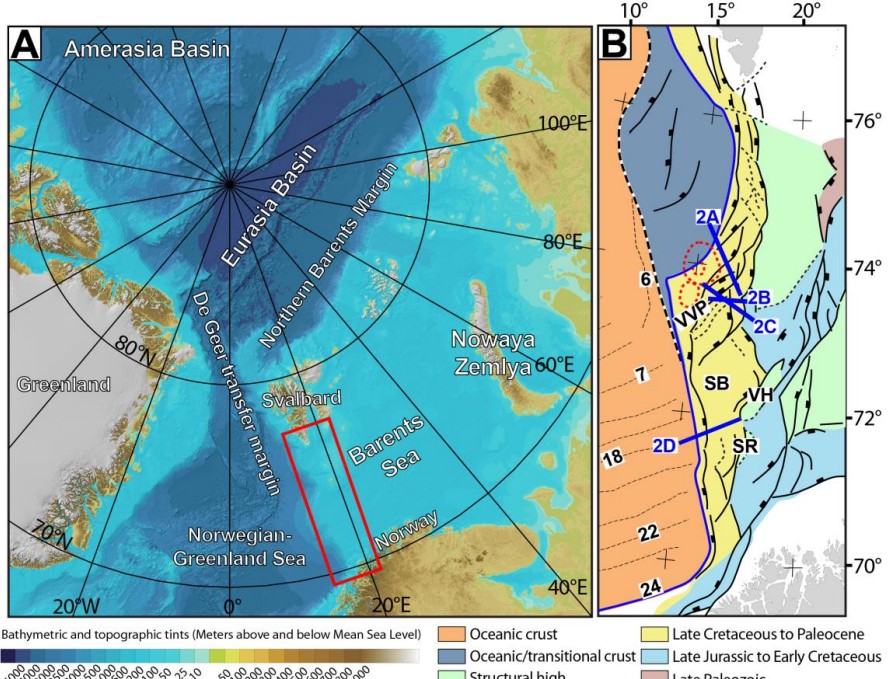

Figure 1: A) The Barents Sea provides is separated from the Norwegian-Greenland Sea by the de Geer transfer margin. Red box shows the present study area. B) Structural map Barents Sea shear margin. Note segmentation of the continent-ocean transition. Abbreviations: HFZ=Hornsund Fault Zone, KFC = Knølegga Fault Complex, SFZ=Senja Shear Zone, SR=Senja Ridge, SB = Sørvestsnaget Basin, VVP = Vestbakken Volcanic Province.

**Regional background**

In the following sections we provide definitions and a short description of the most important structural elements constituting the study area. All structural elements described below are displayed in **Figure 1B**.

**The Barents Shear Margin** was preceded by the "De Geer Zone" (Eldholm et al. 1987; 2002; Faleide et al. 1988; Breivik et al. 1998; 2003). Together with its conjugate Greenland counterpart it carries the evidence of an extensive period of structuring, starting with post-Caledonian (Devonian) extension and culminating with Cenozoic break-up of the North Atlantic (e.g. Brekke 2000; Gabrielsen et al. 1990; Faleide et al. 1993; Gudlaugsson et al. 1998). Two shear margin segments that are separated by a central rift-dominated segment can be identified in the Barents Shear Margin (Myhre et al. 1982; Vågnes 1997; Myhre & Eldholm 1988; Ryseth et al. 2003; Faleide at al.



1988; 1993; 2008). Each segment maintained a particular signature concerning the
structural and magmatic characteristics of the crust during its development. Of these
the Senja Shear Margin is the southernmost segment, originally termed the Senja
Fracture Zone by Eldholm et al. (1987). Particularly the hanging wall west of the
Knølegga Fault Complex of the Barents Shear Margin was affected by wrench
deformation as seen from several push-ups and fold systems (Grogan et al. 1999; Bergh
& Grogan 2003). Here, NNW-SSE-striking folds interfere with folds with NE-SW-
striking axes (Giennenas 2018). Strain partitioning may also have affected some of the
other shear zone segments of the study area (Kristensen et al. 2017).

**The Senja Shear Margin** was active during the Eocene opening of the Norwegian-
Greenland Sea during dextral shear that was accompanied splitting out slivers of
continental crust that became isolate units embedded by oceanic crust during seafloor
spreading (Faleide et al 2008). The Senja Shear Margin coincides with the western
margin of a basin system that is characterized by significant crustal thinning and
extreme sedimentary thicknesses that may approach 18-20km. This part of the shear
margin was characterized by a composite architecture even at the earliest stages of its
development (Faleide et al. 2008). Subsequently shearing contributed to the
development of releasing and restraining bends, associated pull-apart-basins, neutral
strike-slip segments, flower-structures and fold-systems (*sensu* Crowell 1974a,b;
Biddle & Christie-Blick 1985a,b; Cunningham & Mann 2007a,b). The structuring of
the margin was complicated by active halokinesis (Knutsen & Larsen 1997;
Gudlaugsson et al. 1998; Ryseth et al. 2003).

**The Hornsund Fault Zone and West Spitsbergen Fold-and Thrust Belt** form the
northernmost segment of the Barents Shear Margin and coincides with the northern
continuation of the De Geer Zone and the Senja Shear Margin. The presently
distinguishable master fault of this system is the Hornsund Fault Zone, which together
with the West Spitsbergen fold-and-thrust-belt provides a classical setting for
transpression and strain partitioning (Harland 1965; 1969; 1971; Lowell 1972;
Gabrielsen et al. 1992; Maher et al. 1997; Leever et al. 2011 a,b). Plate tectonic
reconstructions suggest that the plate boundary accommodated c. 750 km along-strike
displacement and 20-40 km of shortening in the Eocene (Bergh et al. 1997; Gaina et al.
206  2009).




**The Sørvestsnaget Basin** occupies the area east the COT between 71 and 73°N and is
characterized by an exceptionally thick Cretaceous-Cenozoic sequence (Gabrielsen et
al. 1990). To the west it is delineated by the Senja Shear Margin and to the northeast it
is separated from the Bjørnøya Basin by the southern part of the Knølegga Fault
Complex (Faleide et al. 1988). The Senja Ridge coincides with its southeastern border,
whereas the Vestbakken Volcanic Province is situated to its north. An episode of
Cretaceous rifting in the Sørvestsnaget Basin seems to have climaxed in the
Cenomanian-middle Turonian (Breivik et al. 1998) to become succeeded by Late
Cretaceous-Palaeocene fast sedimentation (Ryseth et al. 2003). Particularly the later
stages of the basin development were likely strongly influenced by the opening of the
North Atlantic (Hanisch 1984; Brekke & Riis 1987).  Salt diapirism did also contribute
to structuring of this basin (Perez-Garcia et al. 2013).

**The Senja Ridge** runs parallel to the continental margin and coincides with the western
border of the Tromsø Basin. It is characterized by a N-S-trending gravity anomaly
which are interpreted as buried mafic-ultramafic intrusions which are associated with
the Seiland Igneous Province (Fichler & Pastore 2022). The structural development of
the Senja Ridge has been linked to shear affiliated with the development of the shear
margin (Riis et al. 1986).

**The Knølegga Fault Complex** can be seen as a part of the Hornsund fault system
extending from the southern tip of Spitsbergen (Gabrielsen et al. 1990). It trends NNE-
SSW to N-S and defines the western margin of the Stappen High. The vertical
displacement approaches 6 km. Although the main movements along the fault may be
Tertiary of age, it is likely that it was initiated much earlier. The Tetiary displacement
may have had a lateral (dextral) component (Gabrielsen et al. 1990).

**The Vestbakken Volcanic Province** is the central topic of the present contribution. It
represents the rifted segment of the Senja Shear Margin and links the sheared margin
segments that are situated to the north and south of it and occupies a typical right-double
stepping (eastward) releasing-bend-setting. Prominent volcanoes and sill-intrusions
display significant magmatic activity, and three distinct volcanic evens are
distinguished in the Vestbakken Volcanic Province (Jebsen & Faleide 1998; Faleide et





al. 2008; Libak et al. 2012). The area has been affected by complex tectonics and both
extensional and contractional structures are observed. The Vestbakken Volcanic
Province is delineated towards the east by an extensional top-west fault zone that
parallels the Knølegga Fault Complex). The interior of the Vestbakken Volcanic
Province is dominated by NE-SW-striking extensional faults and associated fault
blocks. Positive structural elements include inverted fault blocks, and wide-angle ($\square$ >
20 km) anticlines (roll-over anticlines?) and domes that are overprinted by faults and
folds with amplitudes and wavelengths on the hundred- and km-scales.
The eastern boundary fault (EBF) is a top-west normal fault with a regional NNE-SSW
strike, consisting of two separate, hard-linked segments. Its northern segment dips
more steeply to the WNW than the southern segment. The total vertical displacement
as measured on the early Eocene level is in the order of 300 msec (450m), and the upper
part of the hanging wall displays a normal drag modified by hanging wall tight anticline
suggesting post early Miocene inversion. Several normal, dominantly NE-SW-striking
NW-facing normal faults transect the hanging wall of the EFB-fault. The Central Fault
(CF) is the largest of those and is hard-linked to the central segment of the EFB-fault is
the largest of those. The Central Fault is the most prominent fault of a NW-SE-striking
fault population that characterizes the entre Vestbakken Volcanic Province. Three
episodes of Cenozoic extensional faulting were identified in the Vestbakken Volcanic
Province: (i) a late Paleocene-early Eocene event, which correlates in time with the
continental break-up in the Norwegian-Greenland Sea, (ii) an early Oligocene event is
tentatively correlated to plate reorganization around 34 Ma activated mainly NE-SW
striking faults and (iii) an extensional Pliocene event. Evidence of volcanic activity
coincide with the first two of these events. The Vestbakken Volcanic Province is
constrained to its east by the eastern boundary fault (EBF in **Figure 1B**), that is a part
of the Knølegga Fault Complex, separating the Vestbakken Volcanic Province from the
marginal Stappen High further to the east. To the south and southeast the Vestbakken
Volcanic Province drops gradually into the Sørvestsnaget Basin across the southern
extension of the eastern boundary fault and its associated faults. To the west and north
the area is delineated by the continent – ocean boundary/transition (marked as COB in
Fig. 4.1). The Vestbakken Volcanic Province includes both extensional and
contractional structures (e.g. Jebsen & Faleide, 1998; Faleide et al., 2008; Blaich et al.,
2017). Cenozoic tectonic activity has left its imprint at the eastern part of the study area.
All other faults in this map are secondary faults, mainly acting as accommodation



structures to the master faults. Starting from the southern part of the area and south of
the well site, a population of secondary faults is expressed as anastomosing faults
traces.

**Reflection seismic data**

The data set of this study includes 2D seismic reflection data from several surveys and
well data in the Vestbakken Volcanic Province. Data coverage is less dense in northern
part of the study area. Typical spacing of seismic lines is 4km. Well 7316/5-1 was used
to correlate the seismic data with formation tops in the study area whereas published
paper based correlations provided calibration and age of each seismic horizon mapped
(e.g. Eidvin et al., 1993; 1998 Ryseth et al., 2003). Three stratigraphic groups are
present in the well; the Nordland Group (473 - 945 m); the Sotbakken Group (945-
3752m) and Nygrunnen Group (3752-4014m) (Eidvin et al., 1993; 1998;
www.npd.com).

**Fold families**

Several folds of regional significance (with axial traces that can be followed along
strike for 2-3 km or more) occur in the Vestbakken Volcanic Province. The folds
commonly are situated in the hanging walls of extensional fault, some of which bear
the characteristics of tectonic inversion. The fold axial traces parallel the fault traces or
are situated in the strike-continuation of such. It therefore seems obvious that the
structural grain, as defined by the thick-skinned master faults strongly influenced the
positions of the subsequent folds.  The continuity of these structures remains obscure
due to spacing of reflection seismic lines, so each fold may include undetected overlap
zones or axial off-sets that have not been detected. The folds were identified on the
lower Eocene, Oligocene and lower Miocene levels. All the mapped folds are either
positioned in the hanging walls of extensional (sometimes inverted) master faults or are
dissected by younger faults with minor throws.

Three basic fold families were identified in the Vestbakken Volcanic Province by
Giennenas (2018): *Fold family 1* (**Figure 2**) consists of gentle to open anticline-syncline
pairs with upright to slightly inclined axial planes, sometimes with shallowly plunging
fold axes and saddle points, so that the folds are not strictly cylindrical.  Folds of this



family strikes dominantly NE-SW to NNE-SSW and are generally situated at some
distance from the master basin margin faults and in the central parts of the Vestbakken
Volcanic Province where larger faults are less abundant. The wavelengths are in the
order of 3-10 km and the amplitudes may reach 400 m. The fold flanks sometimes
display smaller open folds with wavelengths on the hundred-meter-scale that may be
parasitic, whereas the central parts are commonly broken by post-folding steep brittle
normal faults that define separate fault zones that are separate from the major folds
along strike (see Fold family 3 below). *Fold family 2* includes folds with inclined axial
planes with dominantly long NW-limbs and short SE-limbs (**Figure 2**), which are more
common along the basin margin and in affiliation with low-angle intra-basin reverse
faults. These generally have the characteristics of fault-propagation folds and are
positioned in the hanging walls with steep, inverted normal faults. These are
characterized of axial planes with dips of up to 45 deg) and snake-head-geometries
commonly found in the hanging walls of master faults (**Figure 2A, B**). Fold family
includes anticline-syncline pairs with fold axes that are parallel and are situated more
distally tothe eastern boundary fault (EBF) and are found internally in extensional fault
blocks. The fold axes of the latter sub-family have up-right axial planes and are
accordingly generally oriented N-S

**Strike-slip systems and analogue shear experiments**

Shear margins and strike-slip systems are structurally complex and highly dynamic,
meaning that the incipient and mature stages of strike-slip deformation commonly
display a variety of geometries (e.g. Graymer et al. 2007) making it hard to comprehend
the full complexity solely through fieldwork (e.g. Crowell 1962, 1974a,b; Woodcock
& Fischer 1986; Mousloupoulou et al. 2007; 2008). Analogue

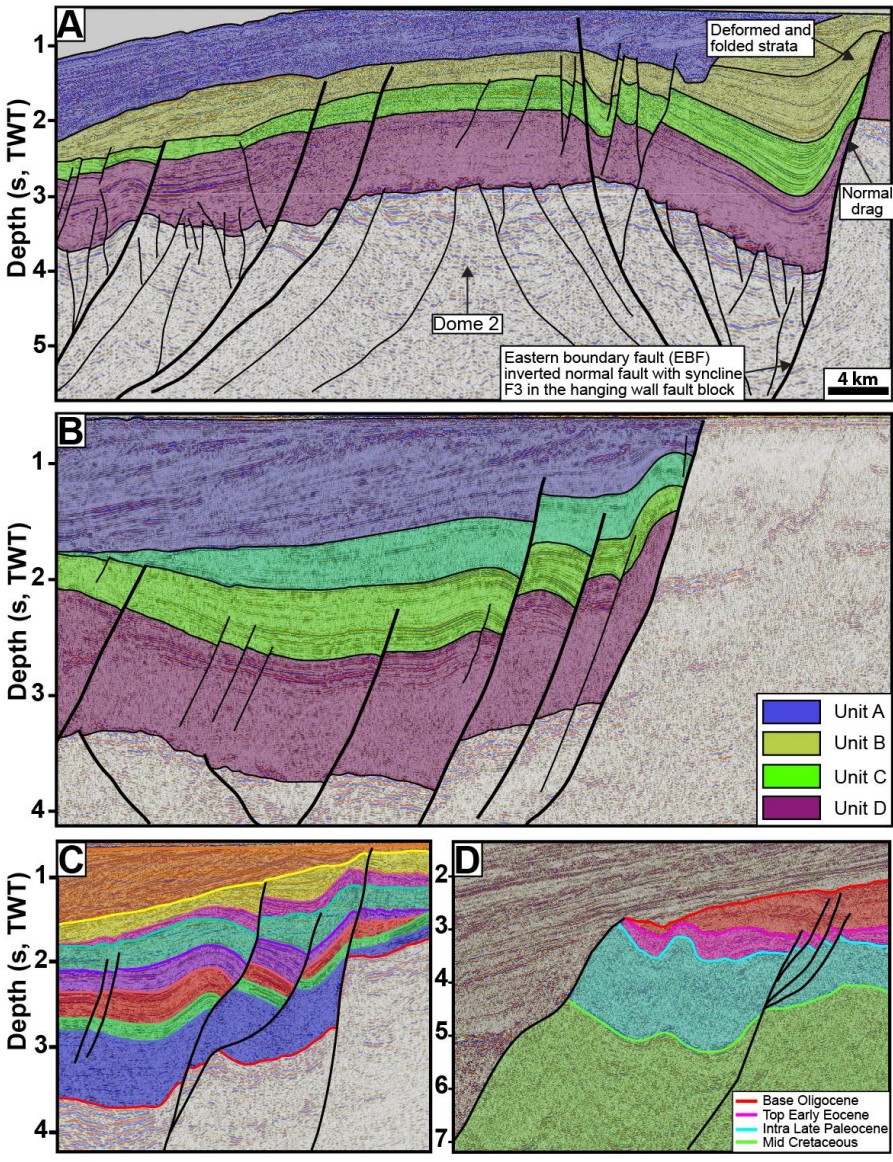

Figure 2: Seismic examples, Vestbakken Volcanic Province. A) Gentle, partly collapsed NE-SW-striking anticline/dome of fold family 1 (Giannenas 2018) of uncertain origin in the eastern terrace domain of the southern Vestbakken Volcanic Province. This may be a remnant, rotated PSE-1-structure (see text for explanation). B,C) Assymetrical folds (fold family 2; Giannenas 2018) situated along the eastern margin of the Vestbakken Volcanic Province. These may represent primary SPE-5-structures focused in the hangingwalls along margins of master fault blocks, or reactivated SPE-2-structures. D) trains of symmetrical folds with upright fold axes (corresponding to PSE-5-structures are preserved inside larger fault blocks. See text for explanation of SPE-structures.



models illustrate such complexity well and therefore attracted the attention of early
workers in this field (e.g. Cloos 1928; Riedel 1929) and have continued to do so until
today. Early experimental works mostly utilized one-layer ("Riedel-box") models (e.g.
Emmons 1969; Tchalenko 1970; Wilcox et al. 1973), which were soon to be expanded
by the study of multilayer systems (e.g. Faugère et al. 1986; Naylor et al. 1986; Richard
et al. 1991; Richard & Cobbold 1989, 1995; Schreurs 1994, 2003; Manduit & Dauteuil
1996; Dateuil & Mart 1998; Schreurs & Colletta 1998, 2003; Ueta et al. 2000; Dooley
& Schreurs 2012). The systematics and dynamics of strike-slip systems have been
focused upon in a number of summaries like Sylvester (1985, 1988); Biddle & Christie-
Blick (1985a,b); Cunningham & Mann (2007); Dooley & Schreurs (2012); Nemcok et
al. (2016) and Peacock et al. (2016). Concepts and nomenclature established in these
works are used in the following descriptions and analysis. Also, following Christie-
Blick & Biddle (1985a,b) and Dooley & Schreurs (2012) we apply the term Principal
Deformation Zone (PDZ) for the junction between the movable polythene plates
underlying the experiment. The contact between the fixed and movable base defined a
non-stationary velocity discontinuity ("VD"; Ballard et al. 1987; Allemand & Brun
1991; Tron & Brun 1991).
Several experimental works have particularly focused on the geometry and
development of pull-apart-basins in releasing bend settings (Mann et al. 1983; Faugére
et al. 1983; Richard et al. 1995; Dooley & McClay 1997; Basile & Brun 1999; Sims et
al. 1999; Le Calvez & Vendeville 2002; Mann 2007; Mitra & Paul 2011). The pull-
apart basin was described by Burchfiel & Stewart (1966) and Crowell (1974a,b) as
formed at a releasing bend or at a releasing fault step-over along a strike-slip zone
(Biddle & Christe-Blick 1985a,b). This basin type has also been termed "rhomb
grabens" (Freund 1971) and "strike-slip basins" (Mann et al. 1993) and is commonly
considered to be synonymous with the extensional strike-slip duplex (Woodcock &
Fischer 1986; Dooley & Schreurs 2012). In the descriptions of our experiments, we
found it convenient to distinguish between extensional strike-slip duplexes in the
context of Woodcock & Fischer (1986) and Twiss & Moores 2007, p. 140-141;) and
pull-apart basins (rhomb grabens: Crowell 1974a,b; Aydin & Nur 1993) since they
reflect slightly different stages in the development in our experiments (see discussion).




## Experimental setup

To study the kinematics of complex shear margins, a series of analogue experiments were performed at the tectonic modelling laboratory (TecLab) of Utrecht University, The Netherlands. All experiments were built on two overlapping 1 mm thick plastic sheets (each 100 cm long and 50 cm wide) that were placed on a flat, horizontal table surface. The boundary between the underlaying movable and overlaying stationary plastic sheets had the shape of the mapped continent-ocean boundary (COB; **Figure 1B**). The moveable sheet was connected to an electronic engine, which pulled the sheet at constant velocity. The modelling material was then placed on these sheets where the layers on the stationary sheet represent the continental crust including the continent-ocean transition (COT) whereas the those on the mobile sheet represents the oceanic crust. The model layers are confined by aluminum bars along the long sides and sand along the short sides (**Figure 3A**). Continental crust tapers off towards the oceanic crust simulating the ocean-continent transition (**Figure 3B**) were included in all models.

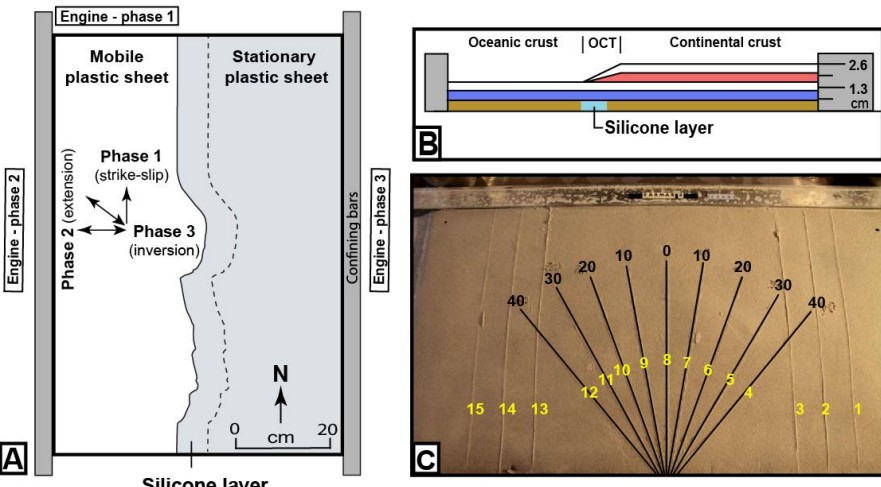

Figure 3: A) Schematical set-up of BarMar3-experiment as seen in map view. B) Section through same experiment before deformation, indicating stratification and thickness relations. C) Standard positions and orientation for sections cut in all experiments in the BarMar-series. Yellow numbers are section numbers. Black numbers indicate angle between the margins of the experiment (relative to N-S) for each profile.



The pre-cut shape of the plate boundary includes major releasing bends positioned so that they correspond to the geometry of the COB and the three main structural segments of the Barents Shear Margin as follows. *Segment 1* of the BarMar-experiments (**Figure 4**) contained several sub-segments with releasing and restraining bends as well as segments of "neutral" (Wilcox et al. 1973; Mann et al. 1983; Biddle & Christie-Blick 1985b) or "pure" (Richard et al. 1991) strike-slip. *Segment 2* had a basic crescent shape, thereby defining a releasing bend at its southern margin in the position similar to that of the Vestbakken Volcanic Province, that merged into a neutral shear-segment along the strike of, whereas a restraining bend occupied the northern margin of the segment. *Segment 3* was a straight basement segment, defining a zone of neutral shear and corresponds to the strike-slip segment west of Svalbard (**Figure 1**).

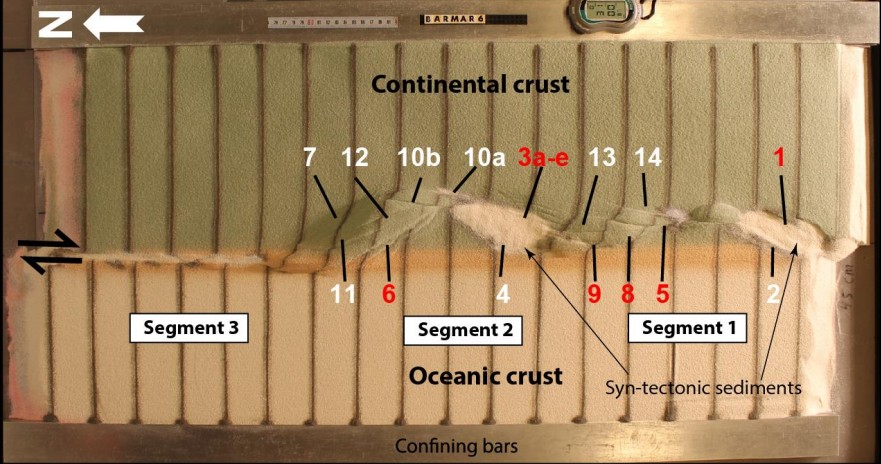

Figure 4: Position of segments and major structural elements as referred to in the text and subsequent figures (see particularly Figures 5 and 6). This example is taken from the reference experiment BarMar6. All experiments BarMar6-9 followed the same pattern, and the same nomenclature was used in the description of these experiments.

The experiments included three stages of deformation with constant rates of movement of the mobile sheet at 10 cmhr$^{-1}$ in all three stages. Dextral shear was applied in the *first phase* in all experiments by pulling the lower plastic sheet by 5cm. In the *second phase* the left side of the experiment was extended by 3 cm orthogonally (BarMar6) or obliquely (325 degrees; BarMar 8 & 9) to the trend of the shear margin, whereas plate motion was reversed during the *third phase of deformation*, leading to inversion of



earlier formed basins that had been developed in the strike-slip and extensional phases.
Sedimentary basins that develop due to strike-slip (phase 1) or extension (phase 2) have
been filled with layers of colored feldspar sand by sieving. These layers are primarily
important for discriminating among deformation phases and thus act as marker
horizons. Phase 3 was initiated by inverting the orthogonal (BarMar6) or oblique
(BarMar 8 & 9,) extension of Phase 2 as a proxy for ridge-push that likely was initiated
when the mid-oceanic ridge was established in Miocene time in the North Atlantic
(Moser et al., 2002; Gaina et al. 2009). Contraction generated by ridge-push has been
inferred from the mid Norwegian continual shelf (Vågnes et al. 1998; Pascal &
Gabrielsen 2001; Faleide et al. 2008; Gac et al. 2016) and seems still to prevail in the
northern areas of Scandinavia (Pascal et al., 2005), although far-field compression
generated by other processes have been suggested (e.g. Doré & Lundin 1996).
Coloured layers of dry feldspar sand represent the brittle oceanic and continental crust.
This material has proven suitable for simulating brittle deformation conditions
(Willingshofer et al., 2005; Luth et al. 2010; Auzemery et al., 2021) and is characterized
by a grain size of 100-200 □m, a density of 1300 kgm$^{-3}$, a cohesion of ~16-45 Pa and a
peak friction coefficient of 0.67 (Willingshofer et al. 2018). Additionally, a 8 mm thick
and of variable width corresponding to the mapped transition zone of 'Rhodorsil
Gomme GSIR' (Sokoutis, 1987) silicone putty mixed with fillers was used as a proxy
for the thinned and weakened continental crust at the ocean-continent transition (**Figure**
**1B and 3A,B**). This Newtonian material (n=1.09) has a density of 1330 kgm$^{-3}$and a
viscosity of 1.42x10$^4$ Pa.s.
The experiments have been scaled following standard scaling procedures as described
by Hubbert (1937), Ramberg (1967) or Weijermars and Schmeling (1986), assuming
that inertia forces are negligible when modelling tectonic processes on geologic
timescales (see Ramberg (1981) and Del Ventisette et al., (2007) for a discussion on
this topic). The models were scaled so that 10 mm in the model approximates c. 10 km
in nature yielding a length scale ratio of 1.00E$^{-6}$. As such, the model oceanic and
continental crusts scale to 18 and 26 km in nature, respectively, which, although slightly
overestimating the most intensely thinned oceanic crust (10-12 km) is in full agreement
with the estimated thickness of the thinned oceanward segment of the continental crust
(30-20 km Breivik et al. 1998).
The brittle crust, dry feldspar sand, deforms according to the Mohr-Coulomb fracture
criterion (Horsfield 1977; Mandl et al. 1977; McClay 1990; Richard et al. 1991;



Klinkmüller et al. 2016), whereas silicone putty promotes ductile deformation and
folding. The geometry applied in the present experiments is accordingly well suited for
the study of the COB in the Barents Shear Margin (Breivik et al. 1998).
When complete, the experiments were covered with a thin layer of sand further to
stabilize the surface topography before the models were saturated with water and cross-
sections that were oriented transverse to the velocity discontinuity were cut in a fan-
shaped pattern (**Figure 3C**). All experiments have been monitored with a digital camera
providing top-view images at regular time intervals of one minute.

All experiments performed were oriented in a N-S-coordinate framework to facilitate
comparison with the western Barents Sea area and had a three-stage deformation
sequence (dextral shear – opening – closure). All descriptions and figures relate to this
orientation. It was noted that all experiments reproduced comparable basic geometries
and structural types, demonstrating robustness against variations in contrasting strength
of the "ocean-continent"-transition zone, which included by a zone of silicone putty
with variable width below a eastward thickening sand-wedge (**Figure 3B**) and changing
displacement velocities.

**Modelling Results**
A series of totally nine experiments (BarMar1-9) with the set-up described above was
performed. Experiments BarMar1-5 were used to calibrate and optimize geometrical
outline, deformation rate, and angles of relative plate movements and are not shown
here. The optimized geometries and experimental conditions were utilized for
experiments BarMar6-9, of which BarMar6 and 8 (and some examples from BarMar9
and are illustrated here, yielded similar results in that all crucial structural elements
(faults and folds) were reproduced in all experiments as described in the text are shown
in **Figure 4**.) It is emphasized that the extensional basins affiliated with the extension
phase (phase 2) became wider in the orthogonal (BarMar6) as compared to oblique
extension experiments (BarMar 8) (**Figures 5 and 6**). Furthermore, the fold systems
generated in the experiments that utilized oblique contraction of 325/145$^0$ (BarMar8-9)
produced more extensive systems of non-cylindrical folds with continuous, but more
curved fold traces as compared experiments with orthogonal extension/contraction
(BarMar6). The fold axes generally rotated to become parallel to the (extensional)



master faults delineating the pull-apart basins generated in deformation stage 1 in
experiments with an oblique opening/closing angle.
Examples of the sequential development is displayed in **Figures 5 and 6**) and
summarized in **Figure 7**.
Elongate positive structural elements with fold-like morphology as seen on the surface
were detected during the various stages of the present experiments. The true nature of
those were not easily determined until the experiments were terminated and transects
could be examined. Such structures included buried push-ups (*sensu* Dooley &
Schreurs 2012), antiformal stacks, back-thrusts, positive flower structures, fold trains,
and simple anticlines. For convenience, we use the non-genetic term "positive structural
elements" termed *PSEm-n* for such structure types as seen in the experiments in the
following description.
In the following the deformation in each segment is characterized for the three
deformation phases.


**Deformation phase 1: Dextral shear stage**

*Segment 1:* Differences in the geometry of the pre-cut fault trace between segments 1,
2 and 3 became evident after the very initial deformation stage. Particularly in segments
1 and 3 an array of oblique *en échelon* folds in between Riedel shear structures (*PSE-*
*1-structures*) oriented c. 145º(NW-SE) to the regional VD rotating into a NNW-SSE-
orientation by continued shear (**Figure 8**; see also Wilcox et al. 1973; Ordonne &
Vialon 1983; Richard et al. 1991; Dooley & Schreurs 2012).  These were simple,
harmonic folds with upright axial planes and fold axial traces extending a few cm
beyond the surface shear-zone described above. They had amplitudes on the scale of a
few millimeters and wavelengths on scale of 5 cm. The PSE-1-structures interfered with
or were dismembered by younger developing structures, also causing northerly rotation
of individual intra-fault zone lamellae (remnant PSE-1-structures (**Figure 8**). Structures
similar to PSE-1-fold arrays are known from almost all strike-slip experiments reported
and described in the literature from the early works of (e.g. Cloos 1928; Riedel 1929.
See Dooley & Schreurs 2012 for summary) and are therefore not given further attention
here. By 0.25 cm of horizontal displacement in segment 1, which included two releasing
and restraining bends in combination with strands of neutral shear, a slightly curvilinear





surface trace of a NE-SW-striking, top-NW normal faultsin the southernmost part of
segment 1 developed. This co-existed with the PSE-1-structures and was immediately
paralleled by a normal fault with opposite throw (fault 2, **Figure 4**) so that the two
faults constrained a crescent- or spindle-shaped incipient extensional shear duplex
(**Figures 5B and 6B**; see also Mann et al. 1983; Christie-Blick & Biddle 1985; Mann
2007; Dooley & Schreurs 2012).
A system of *en échelon* separate N-S to NNE-SSE- striking normal and shear fault
segments became visible in segment 1 after ca. 1 cm of shear (**Figure 5C,D**). These
faults did not have the orientations as expected for R- and R'-shears, but became
progressively linked by alongstrike growth and the development of new faults and fault
segments. They thereby acquired the characteristics of Y-shears, dissecting the PSE-1-
structures). By 2.4 cm of shear, segment 1 had become one unified fault array (**Figure
5D and 6D**), delineating a system of incipient push-ups or positive flower structures
(*PSE-2-structures;* **Figures 8 and Figure 10, sections B1 and B3**, see also; Riedel
1929; Wilcox et al. 1973; Odonne & Vialon 1983; Dauteuil & Mart 1995; Dooley &
Schreurs 2012).
The PSE-2-structures had amplitudes of 1 - 2 cm and wavelengths of 3 - 5 cm as
measured on the surface with fault surfaces that steeped down-section, the deepest parts
of the structures having cores of sand-layers deformed by open to tight folds. The folds
had upright or slightly inclined axial planes, dipping up to 55º, mainly to the east.  The
structures also affected the shallowest layers down to 1-2 cm in the sequence, but the
shallowest sequences were developed at a later stage of deformation and were
characterized by simple gentle to open anticlines.
These structures were constrained to a zone of deformation directly above the trace of
the basement fault, similar to that commonly seen along shear zones (e.g. Tchalenko
1971; Crowell 1974 a,b; Dooley & Schreurs 2012). This zone was 3-4 cm wide and
remained stable throughout deformation stage 1 and  was restricted to the close vicinity
of the basement shear fault itself as also described from one-stage shear faults in Riedel
box-type experiments (e.g. Tchalenko 1970; Naylor et al. 1986; Richard et al. 1991;
Casas et al. 2001; Dauteuil & Mart 1998; Dooley & Schreurs 2012) and from nature as
well (e.g. Wilcox et al. 1973; Harding 1974; Harding & Lowell 1979; Sylvester 1988:
Woodcock & Schubert 1994; Mann 2007).





A horse-tail-like fault array developed by ca. 3 cm of shear at the transitions between
segments 1 and 2 (see also Cunningham & Mann 2007; Dooley & Schreurs 2012, their
Figure 44) (**Figures 5B-D and 6B-D**).

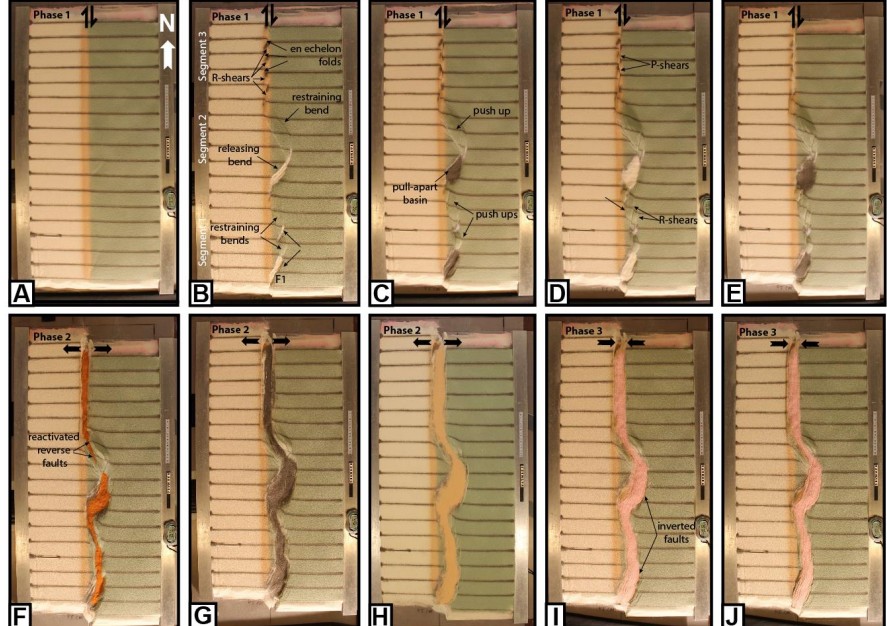


Figure 5: Sequential development of experiment BarMar6 by 0.5, 2.4, 3.5, 4.0 and
5.0cm of dextral shear (Steps A-E), orthogonal extension (steps F-H) and oblique
contraction (steps I-J). The master fault strands are numbered in Figure 4, and the
sequential development for each structural family is shown in Figure 7.

The structuring in *Segment 2;* was ruled by the crescent-shaped basement fault (VD)
that generated   a releasing bend along its southern and a restraining bend along its
northern border (**Figure 11**). The first fault of fault array 3a-e in the southern part of
Segment 2 was activated after c. 0.15 cm of bulk horizontal displacement (**Figure 7**).
It was situated directly above the southernmost precut releasing bend, defining the
margin of crescent-shaped incipient extensional strike-slip duplexes (in the context of
Woodcock & Fischer, 1986, Woodcock & Schubert, 1994 and Twiss & Moores, 2007,
p. 140-141). The developing basin got a spindle-shaped structure and developed into a
basin with a lazy-S-shape (Cunningham & Mann 2007; Mann 2007).



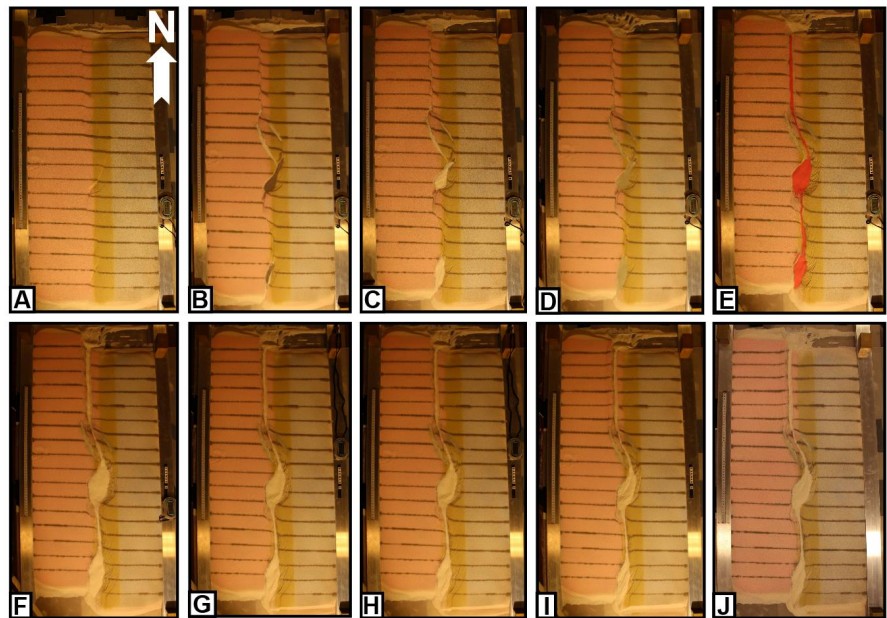


Figure 6: Sequential development of experiment BarMar8 by 0.5, 2.4, 3.5, 4.0 and
5.0cm of dextral shear (Steps A-E), oblique extension (steps F-H) and oblique
contraction (steps I-J). The master fault strands are numbered in Figure 3, and the
sequential development for each structural family is shown in Figure 7. Phases 2 and 3
involved oblique (3250) extension and contraction in this experiment.

The basin widened towards the east by stepwise footwall collapse, generating

sequentially rotating crescent-shaped extensional fault blocks that became trapped as

extensional horses in the footwall of the releasing bend (**Figure 11**). In the areas of the

most pronounced extension the crestal part of the rotational fault blocks became

elevated above the basin floor, generating ridges that influenced the basin floor

topography and hence, the sedimentation. By continued sieving of sand layer on the top

of these structures, forced folds (Hamblin 1965; Stearns 1978; Groshong 1989; Khalil

& McClay 2016) were generated (**Figure 10A**). In the analysis we used the term *PSE-*

*3-structures* for these features.

By a shear displacement of 0.55 cm additional curved splay faults were initiated from

the northern tip of the master fault of fault 3f; **Figure 7**), delineating the northern

margin of a rhombohedral pull-apart-basin (Mann et al. 1983; Mann 2007; Christie-

Blick & Biddle 1985) and with a geometry that was indistinguishable from pull-apart

basins or rhomb grabens affiliated with





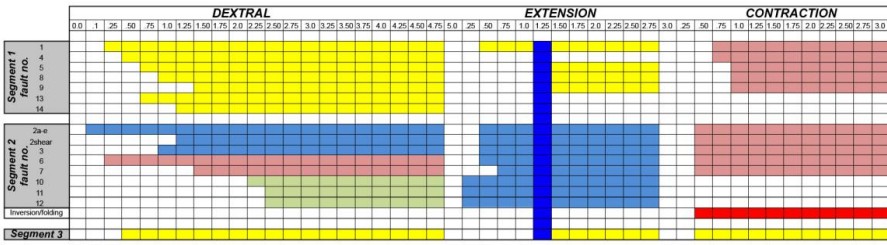


Figure 7: Summary of sequential activity in each master structural element (Figure 4)
in Experiment BarMar6 (Figure 5). Type and amount of displacement is shown in two
upper horizontal rows. The vertical blue bar indicates the stage at which full along-
strike communication became established between marginal basins.

unbridged *en échelon* fault arrays (Crowell 1974 a,b; Aydin & Nur 1993). Although
sand was filled into the subsiding basins to minimize the graben relief and to prevent
gravitational collapse, the sub-basins that were initiated in the shear-stage were affected
by internal cross-faults, and the initial basin units remained the deepest so that the
buried internal basin topography maintained a high relief with several apparent depo-
centers separated by intra-basinal platforms.
Systems of linked shear faults and PSE's became established in the central part with
neutral shear that separate the releasing and restraining bends and development
similarly to that seen for segment 3 (see below), but these structures were soon
destroyed by the combined development of the northern and southern tips of the
extensional and contractional shear duplexes (**Figure 10**).
The first structure to develop in the regime of the restraining bend (segment 2; was a
top-to-the-southwest (antithetic) thrust fault at an angle of $145^0$ with the regional trend
of the basement border as defined by segments 1 and 3 (Fault 6). It became visible by
0.5 cm of displacement. The northern part of segment 2 became, however, dominated
by a synthetic contractional top-to-the-northeast fault that was initiated by 0.85 cm of
shear (Fault 7 **Figures 5 and 6**). Thus, faults 6 and 7 delineated a growing half-crescent-
shaped 5-7-cm wide push-up structure (Aydin & Nur 1982; Mann et al. 1983) south of
the restraining bend (**Figure 9**; *PSE-4-structures*).  By continued shear these structures
got the character of an antiformal stack.
*Segment 3* defined a straight strand of neutral shear. Its development in the BarMar-
experiments followed strictly that known from numerous published experiments (e.g.
Tchalenko 1970; Wilcox et al. 1973; Harding 1974; Harding & Lowell 1979; Naylor et
al. 1986; Sylvester 1988; Richard et al. 1991; Woodcock & Schubert 1994; Dauteuil &





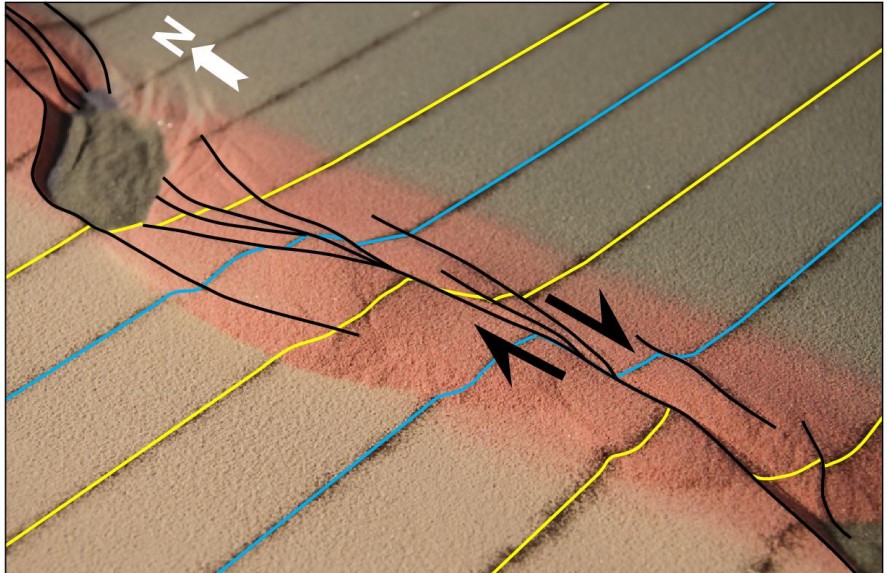

Figure 8: PSE-1 anticline-syncline pairs in segment 1 experiment BarMar6 in an oblique view. PSE-1 folds were constrained to the very fault zone and the fold axes (blue lines) and extended only 3-4 cm beyond the fault zone. PSE-2 structures (incipient push-ups and positive flower structures; yellow lines) were delineated by shear faults and completely cannibalized PSE-1 structures by continued shear. Yellow and blue lines show the rotation of the fold axial trace caused by dextral shearing of c. 1,5 cm. 25mm of dextral shear. By a displacement of 35mm the remains of the PSE-1 structure was completely obliterated. The distance between the markers (dark lines) is 5cm. Yellow arrow marks north-direction. White arrows indicate shear direction.

Mart 1998; Mann 2007; Casas et al. 2001; Dooley & Schreurs 2012). A train of Riedel-shears, occupying the full length of the segment, appeared simultaneously on the surface after a shear displacement of 0.5 cm, occupying a restricted zone with a width of 2-3 cm. The Riedel-shears dominated the continued structural development of Segment 3. Riedel'-shears were absent throughout the experiments, as should be expected for a sand-dominated sequence (Dooley & Schreurs 2012). P-shears developed by continued shear, creating linked rhombic structures delineated by the Riedel- and P-shears generating positive structural elements with NW-SE- and NNE-SSE-striking axes (see also Morgenstern & Tchalenko 1967), soon coalescing to form Y-shears. Transverse sections document that these structures were cored by push-up anticlines, positive half-flower structures. The segments with neutral shear would generate full-fledged positive flower structures in the advanced stages of shear (**Figures 5 and 6**). These were accompanied by the development of *en échelon* folds and flower





structures as commonly reported from strike-slip faults in nature and in experiments.
The width of the zone above the basal fault remained almost constant throughout the
experiments, but was somewhat wider in experiments with thicker basal silicone
polymer layers, similar to that commonly described from comparable experiments (e.g.
Richard et al. 1991).

**Deformation Phase 2: Extension**
The late Cretaceous-Palaeocene dextral shear was followed by pure extension
accompanying the opening along the Barents Shear Margin in the Oligocene. Our
experiments utilized focused on the effects of oblique extension, acknowledging that
plate tectonic reconstructions of the North Atlantic suggest an extension angle of 325º
as the most likely (Gaina et al. 2009).
All strike-slip basins widened in the extensional stage, and most extensively so for the
experiments with orthogonal extension. The widening of the basin enhanced the
topography already generated in the shear-stage in the extensional strike-slip duplex in
segment 2 (PSE-3-structures). In the earliest extensional stage the strike-slip basin in
segment 2 dominated the basin configuration, but by continued extension the linear
segments and the minor pull-apart basins in segments 1 and 2 started to open and
became interlinked, subsequently generating a linked basin system that paralleled the
entire shear margin (**Figures 5F-G, 6F-G**).



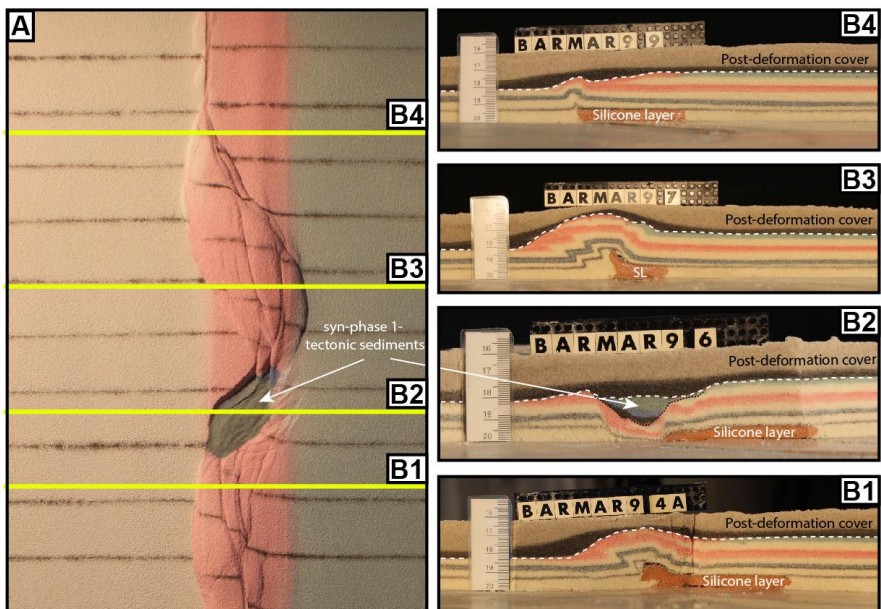


Figure 9: Cross-sections through PSE-2-related structures. A) Folded core of incipient push-up/positive flower structure in segment 1, experiment BarMar6. The fold structure is completely enveloped of shear faults that have a twisted along-strike geometry. Note that the eastern margin of the structure developed into a negative structure at a late stage in the development (filled by black-pink sand sequence) and that the silicone putty sequence (basal pink sequence) was entirely isolated in the footwall.    B) Similar structure in experiment BarMar8. The weak silicone putty layer here bridged the high-strain zone and focused folding that propagated into the sand layers (blue). The folds in upper (pink layers) were associated with the contractional stage, because they contributed to a surface relief filled in by red-black-sand sequence that was sieved into the margin during the contractional stage. C) Contraction associated with "crocodile structure" in the footwall of the main fault in segment 1, experiment BarMar8. Note disharmonic folding with contrasting fold geometries in hanging wall and footwall and at different stratigraphic levels in the footwall, indicating shifting stress situation in time and space in the experiment.D) Transitional fault strand between to more strongly sheared fault segments (experiment BarMar9).

The orthogonal extension-phase following dextral strike-slip reactivated and very quickly linked several of the master faults that were established in deformation phase 1 (**Figures 5A and 6A**) already by an extension of 0,25 – 0,50 cm. This included the southern fault margin, the push-up and the splay faults defining a crestal collapse graben of the push-up (Faults 6, 11 and 12; **Figure 4**). All three segments were reactivated in extension by c. 1.25 cm of orthogonal stretching (**Figure 7**). During the first cm of extension each basin remained an isolated unit, but after 1 cm of extension





all basins became linked, thus forming one unified elongate extensional basin (marked
by the vertical dark blue line in **Figure 7**) and mainly following the PDZ as it was cut
in the basal templates. Among the faults that were inactive and remained so throughout
the extension phase were the antithetic contractional fault delineating the push-ups in
segment 2 towards the south (Fault 6; **Figure 4**). The Y-shear in Segment 3 was
reactivated as a straight, continuous extensional fault in Stage 2. Total extension in
Phase 3 was 5 cm.

## Deformation Phase 3: contraction

In our experiments the extension stage was followed by orthogonal or oblique
contraction (parallel to the direction of extension as applied for each experiment). The
experiments were terminated before the full closure of the basin system, i.e. the
extension vector > contraction vector. A part of the early-stage contraction was
accommodated along new faults. It was more common, however, that faults that had
been generated in the strike-slip and extensional stages became reactivated and rotated.
This was particularly the case for the master faults., as seen by continued or accelerated
subsidence. The dominant structures affiliated with the contractional stage was still new
folds with traces oriented orthogonal to the shortening direction and sub-parallel to the
preexisting master fault systems that defined the margin and basin margins (**Figure 12**).
Also some deep fold sets that had been generated during the strike-slip phase and seen
as domal surface features became reactivated, causing renewed growth of surface
structures (see **Figure 10** and explanation in figure caption). These folds were generally
up-right cylindrical buckle folds in the initial contractional and with very large trace
length: amplitude-ratio (*SPE-5-structures*). Some intra-basinal folds, however, defined
fold arrays that diagonally crossed the basins.  Particularly the folds situated along the
basin margins developed into fault propagation-folds above low-angle thrust planes.
Such faults aligning the western basin margins could have an antithetic attitude relative
to the direction of contraction.
During the contractional phase the margin-parallel, linked basin system started
immediately to narrow and several fault strands became inverted. The basin-closure
was a continuous process until the end of the experiment by 3 cm of contraction. The
contraction was initiated as a proxy for an ESE-directed ridge-push stage.  The first
effect of this deformation stage was heralded by uplift of the margin of the established





shear zone that that had developed into a rift during deformation stage 2. This was
followed by the reactivation and inversion of some master faults (e.g. fault a2; e.g.
**Figure 4**) and thereafter by the development of a new set of low-angle top-to-the-ESE
contractional faults. These faults displayed a sequential development, (fault family 1;
**Figure 4**) and were associated with folding of the strata in the rift structure, probably
reflecting foreland-directed in-sequence thrusting.


## Discussion

The break-up and subsequent opening of the Norwegian-Greenland Sea was a multi-
stage event (**Figure 13**) that imposed shifting stress relation on the already
geometrically complex Barents Shear Margin. The incipient stages occurred in the late
Cretaceous – early Palaoecene (e.g. Eldholm et al 1987; Vågnes 1997; Myhre &
Eldholm 1988; Gabrielsen et al. 1990; Gudlaugsson & Faleide. 1994; Knutsen & Larsen
1997; Ryseth et al. 2003; Faleide at al. 1993; 2008; Reemst et al. 1994; Bergh & Grogan
2003; Kristensen et al. 2017), and the development included extensive volcanism
(Eldholm et al. 1989; Saunders et al. 1997; Planke et al. 1999; Ganerød et al. 2010;
Horni et al. 2017). Opening accelerated and was accompanied by extensive marine
sedimentation in the Eocene and changed into a passive margin setting from the earliest
Oligocene. The spreading stage was likely associated with ridge-push (e.g. Doré &
Lundin 1996; Vågnes et al. 1998; Pascal & Gabrielsen 2001), plate reorganization
(Talwani & Eldholm 1977, Gaina et al. 2009) or other far-field stresses (Doré & Lundin
1996; Lundin & Doré 1997; Doré et al. 1999; Lundin et al. 2013). Faults, and fold
arrays associated with shear and tectonic inversion as well as elements of
subsidence/elevation are obviously prominent for the deduction of the structural history
in such systems (e.g. Cloos 1928, 1955; Riedel 1929; Campbell 1953; Tchalenko 1970;
Wilcox et al. 1973; Dauteuil & Mart 1998; Odonne & Vialon 1983; Richard et al. 1991;
Richard & Kranz 1991; Dooley & McClay 1997; Basile & Brun 1999; Mitra & Paul
2011; Dooley & Schreurs 2012; Kristensen et al. 2017). Therefore, scaled experiments
were designed to illuminate these complexities of the Barents Shear Margin. The
experiments utilized three main segments that correspond to the Senja Fracture Zone
(segment 1), the Vestbakken Volcanic Province (segment 2) and the Hornsund Fault



Zone (segment 3). A series of structural families developed during the experiments,
most of which correspond to structural elements found along the Barents Shear Margin.
Segment 1 in the experiments (which correponds to the Senja Fracture Zone) was
dominated by neutral dextral shear, although subordinate jogs in the (pre-cut) fault
provided minor sub-segments with mainly releasing and subordinate restraining bends.
PSE-1-folds, that developed at an incipient stage were immediately paralleled by two
sets of normal faults with opposite throw in the releasing bend areas (e.g. fault 2 **Figure**
**4**) so that the two faults constrained a crescent- or spindle-shaped incipient extensional
shear duplex became evident (**Figures 5B and 6B**; see also Mann et al. 1983; Christie-
Blick & Biddle 1985; Mann 2007; Dooley & Schreurs 2012). The most prominent of
these structures corresponds to the position of the Sørvestsnaget Basin (**Figure 1B**).
Structures generated at an early stage in a developing, multistage systems that involve
shear, extension and contraction soon became overprinted and canabalized by younger
structures with axes parallel to the main shear fault (Y-shears). The strike-slip stage of
the experiments is comparable to the experiments in series "e" and "f" of Mitra & Paul
(2011).  It is particularly emphasized that SPE-1 and SPE-2-structures were confined
to the area just above the basal master fault (VD) and its immediate vicinity. Although
careful search for positive early (PSE-1) in the reflection seismic data was conducted,
no remains of such structures were detected.
During the oblique extension stage segment 1 of experiments BarMar7-9 were
characterized by oblique opening. The basin subsidence was focused in the minor pull-
apart basins, which soon became linked along the regional N-S-striking basin axis.
Remains of several such basin centers, of which the Sørvestsnaget Basin (Knutsen &
Larsen 1997; Kristiansen et al. 2017) is the largest, are preserved and found in seismic
data (**Figure 1b**). During the experiments a continuous basin system was developed in
the hangingwall side of the master fault, but it is not likely that such a superior basin
system ever existed along the Barents Shear Margin.
In the subsequent inversion stage, fold trains with axial traces parallel (PSE-5-folds) to
the basin axis and the master faults characterized segment 1. Remnants of such folds
are locally preserved in the thickest sedimentary sequences affiliated with the Senja
Shear Margin.

Segment 2, which was underlain by a crescent-shaped discontinuity corresponds to
theVestbakken Volcanic Province and the southern extension of the Knølegga Fault



Complex that is a branch of the southern part of the Hornsund Fault Zone (**Figure 1b**).
The part of the Vestbakken Volcanic Province that was the subject of structural analysis
by Giennenas (2018) corresponds to the southern part of segment 2 in the present
experiments. It is dominated by interfering NNW-SSE- and NE-SW striking fold- and
fault systems in the central part of the basis, whereas N-S-structures are more common
along its eastern margin (**Figure 12A**) (Jebsen & Faleide 1998; Giennenas 2018).
Intra-basinal platforms and complex internal configurations seen in the BarMar-
experiments are common in strike-slip basins (e.g. Dooley & McClay 1997; Dooley &
Schreurs 2012) and are consistent with the structural configuration with intra-basinal
depo-centers within the Vestbakken Volcanic province and also in the Sørvestsnaget
Basin (Knutsen & Larsen 1997; Jebsen & Faleide 1998; **Figure 13**).
The positive structural elements that prevail in *segment 3* are similar to PSE-1 and PSE-
2-structres described for segment 1. The structures affiliated with segment 3 in the
BarMar-experiments correspond well to that seen in the reflection seismic sections
along parts of the Spitsbergen and the Senja shear margins (Myhre et al. 1982). Thus,
the structuring in the segment 3 in the BarMar-experiments followed strictly the pattern
well established for neutral shear in that an array of NW-SE-striking *en echelon* wrench
folds (termed PSE-1-structures in the description above; see Cloos 1928; Riedel 1929;
Tchalenko 1970; Wilcox et al. 1973) first became visible and the development of
Riedel- and P-shears. (R'-shears were subdued as expected for sand-dominated
sequences (Dooley & Schreurs 2012).  Continued shear followed by collapse and
interaction between Riedel and P-shears and the subsequent development of Y-shears
initiated push-up- and flower-structure with N-S-axes (PSE-2) structures that were
expressed as non-cylindrical (double-plunging) anticlines on the surface (e.g.
Tchalenko 1970; Naylor et al. 1986). Structures similar to the PSE-2-structures that
were initiated in the present experiments have previously been reported from similar
experiments with viscous basal layers covered by sand (e.g. Richard et al. 1991;
Dauteuil & Mart 1998), and may have contributed to the more complex fold systems
along the Knølegga Fault Complex.
The Knølegga Fault Complex occupies a km-wide zone. The master fault strand is
paralleled by faults with significant normal throws on its hanging wall side and these
are considered to be strands belonging to the larger Knølegga Fault Complex (EBF;
Eastern Boundary Fault; Giennenas 2018; **Figure 12A**).  The EBF zone is a top-west
normal fault with maximum throw of nearly 2000 ms (3000 meters). It can be followed



along its strike for more than 60 km and seems to die out by horse-tailing at its tip-
points. The vicinity of the master faults of the Knølegga Fault Complex locally display
isolated elongate positive structures constrained by steeply dipping faults. These
structures sometimes display internal reflection patterns that seem exotic or suspect in
comparison to the surrounding sequences. Some of these structures resemble positive
flower structures or push-ups or define narrow anticlines. They are found in both the
footwall and hanging wall of the border faults and strike parallel to those and the axes
of these structures parallel the master faults. The traces of such structures can be
followed over shorter distances than the master faults, and do not occur in the central
parts of the Vestbakken Volcanic Province. We speculate that these are rare fragments
of dismembered EPS-2-type structures.
Due to the right-stepping geometry during dextral shear in segment 2, the southern and
northern parts were in the releasing and restraining bend positions,

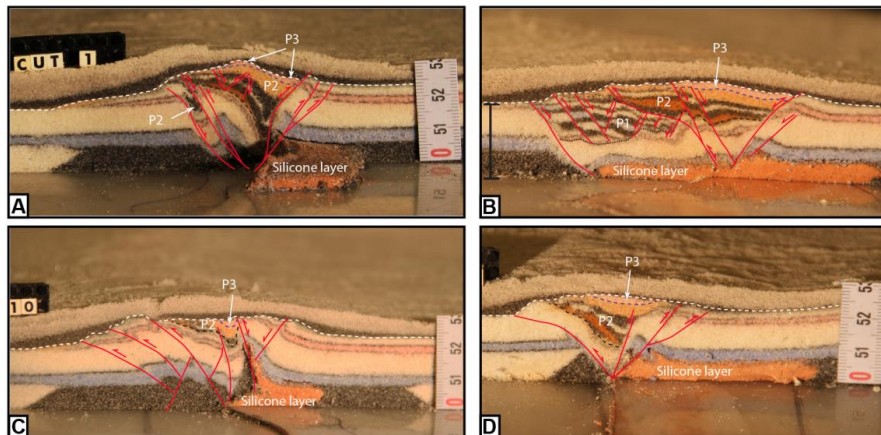


Figure 10: A) contrasting structural styles along the master fault system in segment 2
in map view and (B) cross sections of experiment BarMar9. SL denotes silicone layer,
the stippled line the boundary between pre-and syn-deformation layers and the white
dashed line the boundary with the post-deformation layers.

respectively (e.g. Christie-Blick & Biddle 1985). Hence, the southern part of segment
2 was subject to oblique extension, subsidence and basin formation when the northern
part was subject to oblique contraction, shortening and uplift. The southern segment
expanded to the east and northeast by footwall collapse and activation of rotating fault
blocks that contributed to a basin floor topography that affected the pattern of sediment
accumulation (**Figure 9A, B**). The crests of the rotating fault blocks are termed PSE-3-



structures above, and such eroded fault block crests are defining the footwalls of major
faults in the Vestbakken Volcanic Province, providing space for sediment accumulation
in the footwalls. The area that was affected by the basin formation in the extensional
shear duplex stage seems to have remained the deepest part of the Vestbakken Volcanic
Province, whereas the part formed in basin widening by sequential footwall collapse
created a shallower sub-platform (*sensu* Gabrielsen 1986) (**Figure 11**). It is expected
that (regional) basin and (local) fault block subsidence became accelerated during phase
2 (extension), and more so in the orthogonal extension experiments (BarMar 6) than in
the experiments with oblique extension (BarMar 8), but due to stabilization of basins
by infilling of sand, this was not documented. The widening occurred mainly by fault-
controlled collapse of the footwalls, and dominantly along the master faults that
corresponded to the Knølegga Fault Complex, but also new intra-basinal cross-faults
that were initiated in the shear stage (see above) became reactivated, contributing to the
complexity of the basin topography.

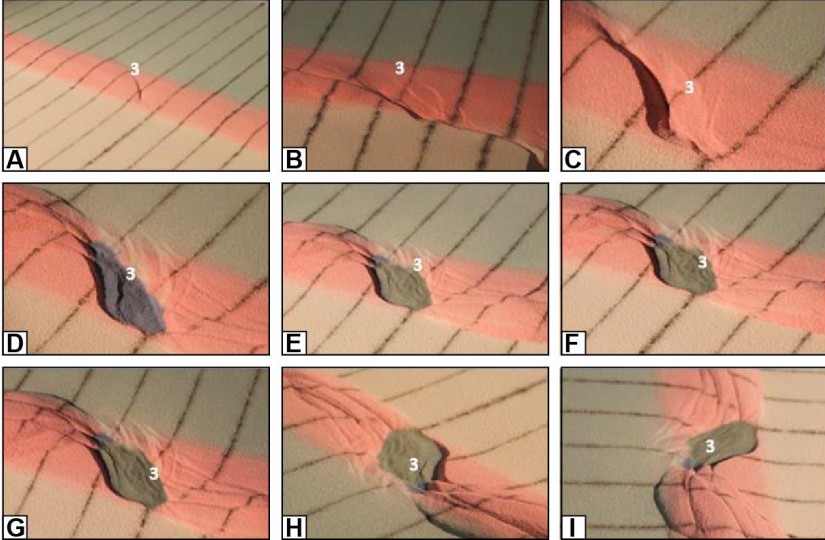


Figure 11: Nine stages in the development of the extensional shear duplex system above
the releasing bend in experiment BarMar9. The master faults that developed at an
incipient stage (e. Fault 3 that constrained the eastern margin of the extensional shear
duplex) remained stable and continued to be active throughout the experiment (Figure
7), but became overstepped by faults in its footwall that became the basin contraction
faults at the later stages H and I. Note that the developing basement was stabilized by
infilling of gray sand during this part of the experiment. Note that Fault 3 remained
active and broke through the basin infill also after the basin infill overstepped the
original basin margin. The distance between the markers (dark lines) is 5cm. Yellow
arrow marks north-direction.



Referring the reflection seismic data from the Barents Shear Margin, it is not likely that a stage was reached where all basins along the margin became fully linked, although sedimentary communication along the margin is likely.

The contraction (phase 3) clearly reactivated normal faults, probably causing focusing of hanging wall strain and folding, rotation of fault blocks and steepening of faults. This means that both intra-basinal and marginal faults in the Vestbakken Volcanic Province can have suffered late steepening. Contraction expressed as fold systems with fold axes paralleling the basin margins development seems to correspond very well to the observed structural configuration of the Vestbakken Volcanic Province. Here pronounced tectonic inversion is focused along the N-S-striking basin margins and along some NE-SW-striking faults in the central parts of the basin. Pronounced shortening also occurred inside individual reactivated fault blocks either by bulging of the entire sedimentary sequence or as trains of folds (**Figure 12**).

The restraining bend configuration in the northern part of segment 2 was characterized by increasing contraction across strike-slip fault strands that splayed out to the northwest from the central part of segment 2 in an early stage of dextral shear.. This deformation was terminated by the end of phase 1 by

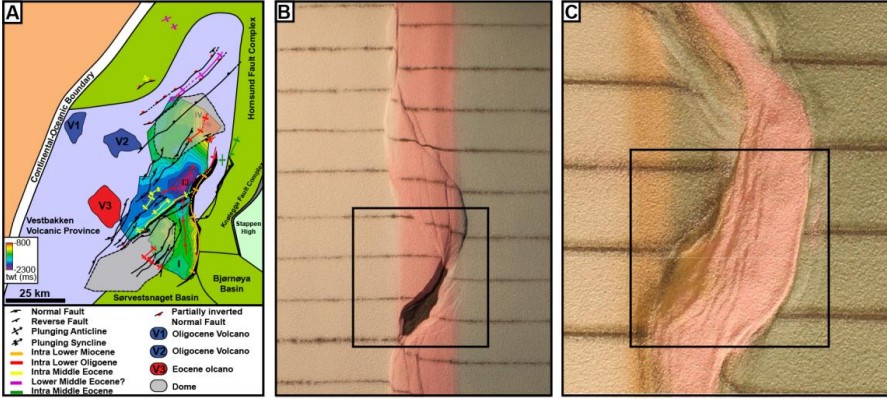

Figure 12: PSE-5-folds generated during phase 3-inversion, experiment BarMar8. Note that fold axes mainly parallel the basin rims, but that they deviate from that in the central parts of the basins in some cases. Note that the folds are best developed in segment 2, which accumulated extension in the combined shear and extension stages.

stacking of oblique contraction faults (PSE-4-structures), defining and antiformal stack-like structure. This type of deformation falls outside the main area, but to the



north this type of oblique shortening during the Eocene (phase 1) was accommodated
by regional-scale strain partitioning (Leever et al. 2011a,b).
The Vestbakken Volcanic Province is characterized by extensive regional shortening.
Onset of this event of inversion/contraction is dated to early Miocene (Jebsen & Faleide
1998, Giennenas 2018) and this deformation included two main structural fold styles.
The first includes upright to steeply inclined closed to open anticlines that are typically
present in the hanging wall of master faults. These folds typically have wavelengths in
the order of 2.5 to 4.5 kilometers, and amplitudes of several hundred meters. Most
commonly they appear with head-on snakehead-structures and are interpreted as buckle
folds, albeit a component shear may occur in the areas of the most intense deformation,
giving a snake-head-type geometry. The second style includes gentle to open anticline-
syncline pairs with upright or steep to inclined axial planes open anticlines-synclines
with wavelengths in the order of 5 to 7 kilometers and amplitudes of several tens of
meters to several hundred meters. We associate those with the PSE-4-type structures as
defined in the BarMar-experiments, where folds of the former type are situated in
positions here sedimentary sequences have been pushed against buttresses provided by
master faults along the basin margins, whereas the latter type was developed as fold
trains in the interior basins, where buttressing against larger fault walls was uncommon.
Also this pattern fits well with the development and geometry seen in the BarMar-
experiments, where folding started in the central parts of the closing basins before
folding of the marginal parts of the basin. In the closing stage the folding and inversion
of master faults remained focused along the basin margins.
The experiments clearly demonstrated that contraction by buckle folding was the main
shortening mechanism of the margin-parallel basin system generated in phase 2
(orthogonal or oblique extension) in all segments. In the Vestbakken Volcanic Province
segments of the Knølegga Fault Complex, the EBF and the major intra-basinal faults
contain clear evidence for tectonic inversion, whereas this is less pronounced in others.
The hanging wall of the EBF is partly affected by fish-hook-type inversion anticlines
(Ramsey & Huber 1987; Griera et al. 2018) (**Figure 2D, E**), or isolated hanging wall
anticlines or pairs or trains of synclines and anticlines  (e.g.; Roberts 1989; Coward et
al. 1991; Cartwright 1989; Mitra 1993; Uliana et al. 1995; Beauchamp et al. 1996;
Gabrielsen et al. 1997; Henk & Nemcok 2008), the fold style and associated faults
probably being influenced by the orientation and steepness of the pre-inversion fault
(Williams et al. 1989; Cooper et al. 1989; Cooper & Warren 2010).






945              Figure 13: Main stages in opening of the North Atlantic.





Some structures of this type can still be followed for many kilometers having consistent
geometry and attitude. These structures have not been much modified by reactivation
and are invariably found in the proximal parts footwalls of master faults, suggesting
that these are inversion structures correlate to EPS-type 5-structures in the experiments
developed in areas of focused contraction along pre-existing fault scarps during
Oligocene inversion.
Trains of folds with smaller amplitudes and higher frequency are sometimes found in
fault blocks in the central part of the Vestbakken Volcanic Province (**Figure 12F**).
Although these structures are not dateable my seismic stratigraphical methods (on-lap
configurations etc.) we regard these fold strains to be correlatable with the tight folds
generated in the inversion stage in the experiments (EPS-5-structures) and that they are
contemporaneous with the EPS-type5-structures.
Segment 1 in the experiments that corresponds to the Senja Shear Margin displays a
structural pattern that is a blend done between configurations done in segments 1 and
2, and the general conclusions drawn above are valid for this part of the shear margin.

**Summary and conclusions**
The Barents Shear Margin is a challenging target for structural analysis both because it
represents a geometrically complex structural system with a multistage history, but also
because high-quality (3D) reflection seismic data are limited and many structures and
sedimentary systems generated in the earlier tectonothermal stages have been
overprinted and obliterated by younger events. This makes analogue experiments very
useful in the analysis, since they offer a template for what kind of structural elements
can be expected. By constraining the experimental model according to the outline of
the margin geometry and imposing a dynamic stress model in harmony according to
the state-of-the-art knowledge about the regional tectono-sedimentological
development, we were able to interpret the observations done in reflection seismic data
in a new light.

Our observations confirmed that the main segments of the Barents Shear Margin, albeit
undergoing the same reginal stress regime, display contrasting structural configurations



The deformation in segment 2 in the BarMar-experiments, was determined by releasing
and restraining bends in the southern and northern parts, respectively. Thus, the
southern part, corresponding to the Vestbakken Vocanic Province, was dominated by
the development of a regional-scale extensional shear duplex as defined by Woodcock
& Fischer (1983) and Twiss & Moores (2007). By continued shear the basin developed
into a full-fledged pull-apart basin or rhomb graben (Crowell 1974; Aydin & Nur 1982)
in which rotating fault blocks were trapped. The pull-apart-basin became the nucleus
for greater basin systems to develop in the following phase of extensional and also
provided the space for folds to develop in the contractional phase.

We conclude that fault- and fold systems found in the realm of the Vestbakken Volcanic
Province are in accordance with a three-stage development that includes dextral shear,
(oblique) extension and contraction along a shear margin with composite geometry.
Folds with NE-SW-trending fold axes that are dominant in wider area of the
Vestbakken Volcanic Province and are dominated by folds in the hanging walls of
(older) normal faults, sometimes characterized by narrow, snake-head- or harpoon-type
structures that are typical for tectonic inversion (Cooper et al. 1989; Coward 1994;
Allmendinger 1998; Yameda & McClay 2004; Pace & Calamitra 2014) typical of
inverted faults.

Comparing seismic mapping and analogue experiments it is evident that a main
challenge in analyzing the structural pattern in shear margins of complex geometry and
multiple reactivation is the low potential for preservation of structures that were
generated in the earliest stages of the development.



**Code/Data availability**

The seismic data that support the findings of this study are available from NPD and TGS. Restrictions apply to the availability of these data, which were used under license for this study.

**Author contribution**

R.H.Gabrielsen: Contributions to outline, design and performance of experiments. First writing and revisions of manuscript. First drafts of figures.

P.A.Giennenas: Seismic interpretation in the Vestbakken Volcanic Province. Identification and description of fold families.

D.Sokoutis: Main responsibility for set-up, performance and handling of experiments. Revisions of manuscript.

E.Willigshofer: Performance and handling of experiments. Revisions of manuscript. Design and revisions of figure material.

M.Hassaan:  Background seismic interpretation. Discussions and revisions of manuscript. Design and revisions of figure material.

J.I.Faleide: Regional interpretations and design of experiments. Participation in performance and interpretations of experiments. Revisions of manuscript, design and revisions of figure material.

**Competing interests**

At least one of the (co-)authors (Ernst Willingshofer) is a guest member of the editorial board of Solid Earth for special issue (Analogue modelling of basin inversion). To the best of our knowledge, no other conflict of interest, financial or other, exists.

**Acknowledgements**

The study is supported by the ARCEx (Research Centre for Arctic Petroleum Exploration) and Trias North projects which are funded by the Research Council of Norway (grant number 228107 and 34152 respectively) together with several academic and industry partners. We want to thank all academic institutes, industry and funding partners. Muhammad Hassaan and Jan Inge Faleide was additionally financed by the Suprabasins project (Research Council of Norway grant no. 295208). We also acknowledged support staff of the Tectonic Modelling laboratory ("TecLab") at the Utrecht University, Netherlands for providing help to perform the analog experiments.



Schlumberger is thanked for providing academic licenses to the Petrel© software. The
Norwegian Petroleum Directorate (NPD) and TGS-NOPEC Geophysical Company
ASA are also acknowledged for providing access to the regional 2D seismic data. The
technical contents and ideas presented herein are solely the authors' interpretations.




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
