# Peer review of "Analogue experiments on releasing and restraining bends and their application to the study of the Barents Shear Margin"

_EGUsphere, 2023_

## Referee Comment (RC1)

**Review of "Analogue experiments on releasing and restraining bends and their application to the study of the Barents Shear Margin"**

**Summary**:

  This paper uses analog modeling to address the structural and basinal evolution of the Barents shear margin. The article has the scientifically significant potential to help understand both this topic and contribute to discussions on basin inversion and how plate margins evolve through varying deformation modes.

  The manuscript begins with a regional background and description of locations of interest and their major structures. This section provides important information; however, grammatical mistakes and missing elements in Figure 1 detract from its utility.

  The following section enters a detailed description of the models and structural observations made in the models. The models follow a velocity discontinuity approach using a margin cut to match the Barents shear margin. This section is generally okay, yet the authors could include additional details about the methodology and model scaling. For example, dimensional considerations of each layer, specifically the dimensions of the silicone strip and its role in nucleating deformation, are only briefly mentioned. Furthermore, analyses of these models comprise only visual analysis of photos taken during the evolution and cross-sectional analysis. These experiments are lacking in that they could have used more state-of-the-art analysis techniques like particle image velocimetry to better understand the fault kinematics and deformation partitioning throughout the model's development. The figures in this section also seem out of place/mixed up, and poorly labeled. Yet, ultimately, the experiments overall reproduced the major basins and structures seen in the Barents shear margin, allowing the authors to conclude that the sequence of phases supports the established tectonic models.

  My main reservation is that in the manuscript, comparison and discussion of the model and natural case are largely descriptive and sometimes repeat things already established in the introduction. There is little discussion about what we can learn about the process. What is the role of natural conditions that cannot be included in the model? For example, the thermal structure of the volcanic province. What explains the differences between the models and nature? What are the more significant takeaways? How does basins stratigraphy link to margin evolution? That said, the authors explain some critical insights, including strain localization and the changing geometry of marginal faults in the Vestbakken Volcanic province, the timing of basin linkage, and constraints on the shortening mechanisms of the margin parallel basin systems. If the authors included more interpretations of the relationship between model results and the evolution of the Barents margin, the paper would be much more substantial. And, if they used PIV, certainly more could be learned.

  One glaring issue with this paper is the significant spelling and grammar mistakes. These strongly detract from how effectively the material is communicated, including issues with figures that make them challenging to critique and understand (missing scale bars, missing labels, incorrect captions).

**Specific comments:**

Lines 148 – 151: This paper helps understand multistage sheared passive margins. By using analog models, you can investigate the stages of deformation on the way to the present-day setting. However, there do not seem to be any specific questions that the authors are trying to answer. What about the structural complexity? What makes it complex? How well do we understand these

margins at present? Complexity is a blanket term. What exactly don't we know? What are the implications of this work, both specific to the Barents margin and from a process-oriented perspective? The language the authors use is sometimes too vague: set into a dynamic context, complexity, structural elements? These words help frame the main issues, but the authors could expand the ideas to more specific scientific challenges.

Figure 1: Panel B does not include many labels in the caption. The locations of interest are difficult to find without prior knowledge of the area. What do the red dotted lines in this figure represent? What is the dashed line by 6? What do the numbers mean? The figure does not stand alone.

Line 163: It is hard to know where these elements are displayed if there are no labels in Figure 1B

Line 165: This is somewhat confusing language. Preceded by? Did this zone come before, or did the term used to be the *De Geer Zone*?

Lines 167, 193, 219, and many other places: structuring is not a common word and seems strange here.

Line 175: This could be moved to the Senja Shear Margin paragraph.

Line 188: What is meant by "may approach 18 – 20 km"? Are there poor constraints? Or does it only approach that in some locations?

Line 208 – 219: Personally, it makes more sense to describe the Sørvestsnaget basin last since the description includes structural elements that are not introduced until later.

Line 263: Did this volcanic activity affect the crustal evolution of the Barents sea? Did thermal weakening play any modifying role in strain partitioning? Perhaps, this can explain some differences between nature and the models.

Line 273: This comes off as a random or fragmented idea. What is this sentence referring to? The secondary faults? What is the imprint? I think this sentence can be deleted.

Line 294: The characteristics of tectonic inversion? What characteristics? This sentence is too general. Are you referring to lines 295 – 297? Why not say something more specific like: " The axial traces of folds and the structural grain of thick-skinned master faults are roughly parallel. This observation suggests that the position and orientation of folds were influenced by the preexisting structural fabric created by thick-skinned master faulting."

Lines 330 – 331: This is a bit ambiguous. What are the specific limitations? Is there poor exposure or not enough data? Is it hard to comprehend or impossible? I suppose you are saying that you cannot go back in time and see all of this. Is it not more than geometries but variable kinematics and dynamics through time?

Figure 2: Please define the PSE and SPE acronyms here. That would make things easier for the reader than combing the text to find what they mean. Additionally, I think this figure would benefit from more labels, like the snake head geometries.

Line 382: Is the length scaling of the boundary cut here in agreement with the total displacement and rate? i.e., do the convergence rates and amounts scale as well?

Line 389: How was the crust tapered? How was the taper angle chosen? Randomly? Or related to what is observed?

Figure 3: What do the colors represent in panel B? A legend would be helpful here. Is the silicon layer just a strip over the VD?

Lines 405 – 413: These are results and would perhaps be better placed in that section.

Line 425: It would be worth mentioning how the authors chose the obliquity angle. Is there field data that suggests this is on the order of what happened in nature? Trans-pression/-tension experiments indicate that this angle plays a significant role in the structural evolution of the system, so it is alarming that this is glossed over.

Line 429: Apologies because I have never used a sieve in analog models. But, I am unsure what or how sieving works to fill basins or what is being done. Could you explain this more?

Lines 443 – 447: How were the dimensions of the silicon strip chosen? Width, thickness? Was this by trial and error? From my experience with similar experiments, the dimensions of the strip can strongly affect the nucleation and propagation of deformation.

Line 467 – 468: Why did the authors not use PIV in this study? PIV is an excellent and easy-to-use tool for studying analog fault kinematics. The photos taken would work easily with PIV software and be especially useful with marker particles. This technique would greatly expand the dataset and provide much more insight into the behavior of the faults throughout the model evolution.

Lines 499 – 501: This sentence seems unnecessary. Clearly, cross sections must be cut to understand folds and faults in detail.

Line 504: I appreciate the use of non-genetic terminology. However, I think the authors should define PSE, SPSE, and EPS acronyms together.

Line 534: en echelon what? Folds? Faults?

Lines 536, 538: I understand that R, R', P, and Y have been used extensively in the literature for wrench systems. However, it does not hurt to add a sentence or two to remind the reader what their orientations are, at least, especially since the terminology is not always consistent. For example, Y-shears vs. $R_1'$ shears. It just becomes confusing.

Figure 5: This figure is nice and reasonably well-labeled. Still, it should include all the major structural elements described here. There is a measuring tape here, but it cannot be seen due to the low resolution of the pictures.

Figure 6: This figure is a bit too bare to be useful. Where are the labels? Which phase is which? The faults could be labeled too. At the least, it should be consistent with figure 5. There is a measuring tape here, but it cannot be seen due to the low resolution of the pictures.

Lines 592 – 594: Can this be expanded? So, adding the sand forced the structures in the system? Perhaps, this strongly impacts the subsequent model evolution. What if no sand is added? What if less sand is added? What if sand is added syntectonically?

Figure 7: What do the colors mean here?

Line 650 – 651: It would be easier to see these Y shears if labeled in the figures.

Line 652 – 654: Would they? Is this an interpretation or a fact? Is there a reference that can be included here or more explanation?

Line 657 – 659: Again, how were the dimensions of the silicone calibrated?

Figures 9 & 10: I think the captions have been accidentally swapped.

Line 694: Very quickly? In what frame of reference is this fast? Relative to other experiments? Is it scaled to nature?

Line 713: Why was this choice made?

Line 744: What is a "shifting stress relation"?

Lines 745 – 756: I appreciate a summary here and there, but this is quite repetitive.

Line 762 – 763: These "complexities" regarding process or mechanics are incredibly vague. Could this be expanded to be more specific?

Lines 777 – 782: This is very descriptive, but there is little discussion of interpreting these structures in the context of the Barents shear margin. Except at the end, where it is made clear that none of these are seen. Why is this? Perhaps, that in itself provides a clue to the margin's evolution. Maybe this could be investigated more.

Lines 785 – 786: This seems a bit redundant. Of course, oblique opening would occur during oblique extension, right?

Lines 790 – 792: Here is another example where there might be something more to say about the Barents margin, but the discussion is cut short. Why was there no superior basin system? Was there not enough extension? The two paragraphs here just say, "we observed something in the model, but we don't actually see it in nature."

Lines 798 – 800: It could be nice to write that on a label showing the segments. Perhaps, written in italics or something so that the reader can look at the fig and completely understand what it represents.

Line 827: What about the complex fold systems? It is still unclear what questions are trying to be answered in this study except to simulate the margin and redescribe the observed structures. There is much more to be said about the structural and basinal history of the margin evolution.

Line 844: What is an EPS-2 structure? Can we see this somewhere in the models (labeled figures)?

Figure 10: It is clear that the captions are mixed up. However, it could be nice here to label the crocodile structures.

Figure 11: Where are the yellow arrows? No scale bar!?

Lines 884 – 887: This paragraph seems fragmented.

Lines 887 – 890: This is a nice, clear, well-described insight provided by the experiments that helps understand the natural system. I think including more discussions like this would make this a stronger manuscript.

Figure 12: Missing scale bar. Nonetheless, this figure nicely highlights things that can be learned from the models about the system's kinematics.

Lines 932 – 943: Another excellent example of where the authors have nicely used the experimental results to provide valuable insights into nature.

Line 972 – 973: What is this new light? Can this be explained more?

Lines 975 – 986: One of the paper's main takeaways: the system's evolution was confirmed by the experimental simulation of deformation events. In itself, this is nice!

Line 990: Were you able to confirm whether the boundary was likely oblique or not? I think it could be emphasized how the obliquity changed the model results and what this means in nature.

**Technical comments:**

Line 185: isolate → isolated

Line 190: Subsequently → subsequent

Line 198: coincides → coincide

Line 201: A classical setting for transpression would be better worded as a "type setting" or "well-described setting."

Line 217: "likely strongly" is poor grammar.

Line 294-295: The grammar in this sentence should be corrected.

Line 308: strikes → strike

Line 323: tothe → to the

Line 387: Whereas  those

Line 480: A series of totally nine?? This paper should be better proofread for English fluency and spelling. → "Nine total experiments" or a "series of nine experiments".
Line 528: faultsin → faults in

Line 544: cm → cms

Line 545: steeped → steepened

Line 664:

Line 716: faults., → faults,

Line 773 – 774: Poor grammar makes the sentence unintelligible.

Line 777: Poor grammar
Lines 814 – 819: Sentence grammar makes it unreadable.

Lines 923 – 927: This sentence is tough to understand.

Line 954: my → by

---

## Referee Comment (RC2)

This manuscript presents seismic profiles and analog modeling to discuss the formation of the Barents Shear Margin. Reviewing such a long manuscript really tasks me, especially that the poor preparation of figures. I cannot find geological units, such as Senja shear margin, Hornsund fault zone, and many other names referred in the text. Even the structures in figure and captions are not matched. Another issue is that too much and redundant description of results prevents me understanding the key evidence and motivation of this work. The text can be shortened by half if they want to have a clearly written paper. I therefore suggest returning this manuscript to authors for a major revision by deleting and combining unnecessary parts, and highlighting the finding or importance of this work.

Below are details of comments:

In the abstract, I do not find any highlighted scientific significance. The last sentence, "this is in general agreement with observations in previous and new reflection seismic data from the Barents Shear Margin", seems to inform us this is only repeating the formation of the rifting margin, while its implications are still in investigation. The uniquity or characteristics of the Barents margin are necessary for readers to know why this problem matters, and thus deserves modeling. This lacks in the abstract.

The Introduction is not informative. Actually, I find one section in lines 327-374 that belongs to "Introduction", because it summarizes previous work and provide motivation to carry out this study. This section is better moved to and incorporated with the current introduction.

As I mentioned above, the major problem bothering me is the mismatched geological unit names and figures. In fact, this is the key to understanding the regional tectonic context. In figure 1, I cannot find KFC, SFZ in the map, while SB and VH are not explained in the captions. Furthermore, there are other units not found on the map, including the Tromso basin, Bjornoya basin, and Stappen High, etc. Please check the background and make sure all the units are easily located on the figures. This is crucial.

The regional background is already of some unnecessary information. Except locating the units, regional tectonic evolution remains untouched. Regional geological history may serve as the connection for these tectonic units by introducing, for example, how many tectonic stages and how magma evolved. With the figure 1, I find I am lost in these complex structures of the Barents shear margin.

In lines 291-302, this whole paragraph presents the fold patterns in the VVP, but neither figures nor references are provided. I do not know if these are based on new data or published data. The purpose of this section is unclear. It belongs to background if identification of fold families comes from Giennenas (2018).

The setup of experiment has 100 lines of words. Some can be moved to supplementary files. Figure 3 shows an uneven margin corresponding to the mapped continent-ocean boundary. I am curious whether this is current or initial shape of continental margin. How can we figure out the initial shape of margin?

I also find figure 3a is different from figure 4. Figure 4 has confining bars on the oceanic crust side, while these bars are to the east of the continental crust of figure 3. I need similar labels of figure 4 in figure 3.

This experiment was conducted with similar rates for all three stages of deformation. I wonder if there are some geological constraints for these rates? Besides, the authors always succeed in writing things in a complex and ambiguous way. Some simple sentences may have better effect than the sentences in lines 423-438. Until the end, I do not know how the engine moves for Phase 3.

Modeling results has 250 lines! In addition to it, there are at least one hundred lines in the discussion which still describe the results of the modeling! These two sections requires reorganization to cut off redundant and repeated description of the observed structures in the sandbox. It seems the authors choose to keep all the details of the experiment, whereas the key results were flooded by them.

My last suggestion is to separate the discussion into several subsections. Tectonic implication deserves its own section.

Line 170: Which two segments?

Line 188: Which basin?

Line 212: Coincides with its border? Temporally or spatially?

Line 225: This sentence is hard to understand.

Line 232: Reference?

Line 246: There is a square.

Line 250: Hard-linked?

Line 271: Fig. 4.1 does not exist.

Line 273: This sentence is not clear. What kind of imprint did Cenozoic tectonics leave?

Line 321: 3 is missed for fold family.

Line 333: Put directions for all profiles. Also, mark the fold types on each figure.

Line 442: 100-200 of what?

Line 449-458: This part is not necessary.

---

## Author Response (AR1)

Dear Editor, Solid Earth

Please find posted our revised version of our article **Analogue experiments on releasing and restraining bends and their application to the study of the Barents Shear Margin.**
We would like to thank the anonymous reviewers for their thorough and insightful reviews, which helped to significantly improve the quality of the manuscript.

We have particularly condensed, simplified and removed some repetitions between introduction and discussion, particularly concerning regional aspects.

Below we detail our response to the reviewers' comments and suggestions.

One annotated and one clean version in word-format of the manuscript that display revisions done by us are posted as separate files.

**Please note that the symbols $\lambda$ and $\mu$m (as seen in word-file lines 246 and 442) comes out as square in the pdf-file that was generated in the editorial procedure.**

**Comments of Reviewer 1:**

Both reviewers find the regional information to be too extensive and repetitive. Several changes have been done to avoid this (see detailed comments related to comments from Reviewer 2) below. Reviewer 1 found the sequence of presentations of structural elements to be non-intuitive. We have restructured this section so that regional structural elements are presented in sequence from north to south.

Details about the **experimental set-up** and dimensions are given in the introduction to the Experiments and in Figure 3B (perhaps overlooked by reviewer?). An addendum D to Figure 3 (photograph) has been produced to highlight the varying width of the silicone putty as described in the text. Expanding the modelling setup description as requested by the reviewer is at variance with the opinion of reviewer 2, who wants us to shorten the modelling setup section.

We agree that using **PIV methods** would allow for a quantitative analysis of the modelling results. Such analysis has previously been performed by us (Leever et al., 2011) in context of strain partitioning in a shear margin environment. Besides that, the reasons why we refrained adding this method are as follows: 1. As PIV software relies on pattern recognition algorithms applied to successive top-view photographs, adding syn-tectonic sedimentation during the model runs, significantly alters the pattern and adds a big deal of uncertainty to the areas where sedimentation has been applied. As such we feel that a qualitative analysis of the models based on top-view and cross-section images allows for a more detailed and accurate description of the models.  2. focusing on the regional aspects of the Barents Shear Margin in this paper, we found that a full PIV-analysis would take much space and attention,

and that this would fit better into an article that would be dedicated to the experimental methods and analysis. For the reasons above we prefer to stick to the model analysis as presented, which in our view is sufficiently well aligned with the available kinematic and geometric data from the natural laboratory.

We have tried to focus the **discussion** on the application of the modelling results on the Barents Sea problems. The discussion and conclusions have been clarified and sharpened on the points requested by reviewer.

**Figures/figure captions** 1, 2, 3, 5, 6, 7, 9 and 10 have been revised according to comments by the reviewer (specified below).

**Grammar and spelling.** The manuscript has been run through word spell- and grammar checks. Misprints and unclear phrasings identified by the reviewer have been corrected. The reviewer has identified some sections with long/complex phrasings. These have been rewritten/simplified.

**General comments from Reviewer 2:**

Abstract has been shortened, omitting general regional information that can be read from the general introduction. Main conclusions have been high-lighted.

Regional descriptions have been modified and simplified to ease discussions of regional structures. Some parts that refer to previous descriptions (like fold families) have been omitted to avoid confusion between present (PSE) and previous nomenclature.

Reference to both seismic interpretations and experimental results have been strengthened and Figure 2 has been expanded to strengthen the relation between observations in reflection seismic data and experiments.

Figures and related text in manuscript have been revised/harmonized to enhance clarity.

Discussion of regional aspects have been simplified on several points.

Reviewer 2 expresses concern about how the geometry of the shear margin (e.g. area of crustal thinning) was constrained. This is fully based on seismic mapping and an additional figure (**Figure 3D**) has been prepared to display the configuration (see comments to figures below).

We have found that taking out lines 449-458 as suggested by Reviewer 2 is at variance with the opinion of Reviewer 1 who wants us to expand on the model setup and scaling section. As these lines contain among others critical information on the length-scale ratio, we did not find it logic to remove this information from the text.

**Specific comments Reviewers 1 and 2**

The following specific changes emphasize points of revisions in line with general revisions as suggested in General comments above. Comments are marked R1 and R2 to link the comment to the review documents in each case.

Lines 148-151 (R1): This section has been expanded and specified when it comes to specification of general problems related to the analyzed area.

Line 165 (R1): The relation between the Barents shear margin and the de Geer zone has been specified.

Lines 167ff (R1): "Structuring" has been changed to "structural development" throughout the manuscript.

Line 170 (R2): Further description of these segments follows in lines 175ff.

Line 175 (R1): Moved to Senja shear margin as suggested.

Line180 (R1): Basin name has been included.

Line 188 (R2): Sedimentary thickness specified.

Line 208ff: (R1) Presentation/description of the Vestbakken Volcanic Province to be moved? We have restructured this section so that regional structural elements are presented in sequence from north to south.

Line 212 (R2): The relative positions of the Sørvestsnaget Basin and the Senja Ridge have been emphasized.

Line 225 (R2): Rephrased

Line 232 (R1): The reference given concerns the two last sentences in the section.

Line 246 (R2): The square in the pdf-file is $\lambda$ in the word document. The l was transformed by the building of the pdf-document.

Line 250 (R1): Changed to "linked"

Line 271 (R2): Misprint.  Figure-reference taken out.

Line 273 (R2): Sentence has been removed.

Lines 294ff (R1): Reformulation as suggested.

Lines 330ff (R1): Introduction to this section rewritten.

Line 331 (R1): Reference to Fold family 3 taken out. (Not used in the following).

Line 389 (R1): Process for determining taper angle included.

Lines 405-413 (R1): Not sure we understand this remark. This describes the experimental set-up and are not results.

Lines 425ff (R1): As described above/below, the angles were chosen according to the analysis of the opening and plate movements by Gaina et al. (2009), see above, the introductory remarks to Descriptions and Discussion. We have added a sentence to underline that here.

Line 429 (R1): Sieve was applied to obtain a smooth surface. Explanation added.

NB: Line 442 (R2): The unit is μm (as seen in word-file). Comes out as square in the pdf-file that was generated in the editorial procedure.

Lines 443-447 (R1): The shape of the silicone putty lid followed the geometry determined from seismic mapping, as described in the following sentence: "Additionally, an 8 mm thick and of variable width corresponding to the mapped transition zone". A parantheses has been added to state this even more clearly.

Lines 449-458 (R2): Information on scaling: See comment under "General comments from Reviewer 2" above.

Lines 467-447 (R1): We have used PIV-methods in previous studies (e.g. Leever et al. 2011a,b). Although favourable for a dedicated experimental work, we feel that the present work mainly addresses regional issues and that the use of PIV-method would seriously expand the text. We still acknowledge this remark and consider writing a pure geometrical version of these results (with some added examples/complexities) to be published separately.

Lines 499-501 (R1: The Reviewer is correct that this may seem obvious. We still feel the need to stress that the positive structures include elements of different types that are not easily distinguished when first detected and monitored thereafter (Figure 7), and that a classification must await the cutting of the experiments.  We have therefore chosen to keep this sentence as it is.

Line 504 (R1): A section has been added here to introduce the reader to the different structure types. We have not used the term "SPSE" as referred to by the Reviewer, but the Reviewer is correct that "EPS" occurs a couple of places in the text. These are misprints for PSE (Positive Structural Element) and have been corrected.

Line 534 (R1): The continuation of the sentence says what kind of structures which was requested by the Reviewer, namely: «a system of *en échelon* separate N-S to NNE-SSE- striking

normal and shear fault segments». We feel that the requested information is given and have therefore kept the sentence as it is.

Lines 536-538 (R1): The full terms and short explanations have been added in parentheses.

Lines 592-594 (R1): The description has been rewritten and expanded.

Lines 650ff (R1): Y-shears have been identified in Figure 8.

Lines 652-654 (R1): This is concluded from the present experiments and documented in Figures 5,6 and 10 as referred in the text.

Lines 657-659 (R1): See explanations to line 443-447 above.

Line 694 (R1): This is just an observation done during the present experiments. We are not aware of other experiments that describes the temporal aspects of fault linkage. We have however determined the process to have taken place between .25 an .50 cm of displacement, and since the velocity of the experiment is known, the timing can be calculated. We did not have any ambition of representing this part of the development in a scaled timeframe, and doubt that this would be scientifically meaningful. We have therefore not changed the text on this point.

Line 713 (R1): The opening velocity is known for the NE-Atlantic (e.g., Gaina et al. 2009). Furthermore, we know that the bulk extension vector was greater than the extension factor by adding up extension and shortening (Vågnes et al. 1997). These references that are referred to elsewhere in the text, have been added here.

Line 744 (R1): Changed to "stress configuration".

Lines 745-763 (R1): The regional summary has been taken out to avoid repetition.

Lines 762-763 (R1): "Structural development" has been substituted for "complexities".

Lines 777-782 (R1): Section expanded and re-written.

Lines 785-786 (R1): Redundant statement obliterated.

Lines 790-792 (R1): The basin linking situation has been specified.

Lines 798-800 (R1): The segments are identified and labelled in Figure 4, so a reference to that figure has been included.

Line 827 (R1): Statement has been reformulated to focus on the influence of mechanically stratified sequences on fold configurations.

Line 844 (R1): This is a misprint. Should read "PSE-structure". Misprint corrected.

Lines 884-887 (R1): Sentence has been simplified.

Lines 972-973 (R1): The section has been re-written and the conclusions have been substantiated.

Line 990 (R1): Statement clarified.

**Figures (additions requested by reviewers)**

**Figure 1.** Missing names (abbreviations) in Figure 1B in map and figure caption have been added/harmonized. We have changed the sequence of abbreviations in the figure captions so that structural elements are presented in sequence from north to south to enhance readability.

**Figure 2:** Seismic examples has been added and abbreviations of structural types (PSE's) have been added. Figure caption has been expanded accordingly.

**Figure 3:** Picture has been added (Figure 3D) and cartoon (Fig. 3A) to display the true geometry of the silicone putty layer.

**Figure 5:** Magnified scale bar has been added.

**Figure 6:** Information has been added for each frame to emphasize key structural elements and configuration/deformation stage. Magnified scale bar has been added.

**Figure 7:** Y-shears and scale (previously given I caption) have been emphasized.

**Table 1:** Table has been added more clearly to identify and explain positive structural element types (PSEs). References to figures that illustrate the different PSE-types have been added.

Figure captions have been revised in harmony with revisions of figures.

**Technical comments from Technical Editor**

References in text have been corrected from format: **"**Author year; Author et al. year" to format: "Author, year; Author et al., year". This has been done throughout the manuscript.

**Additional technical comments (Reviewer 1):**

All technical errors (misprints) pointed to have been corrected

Line 480: Sentence reformulated

Lines 773-774: Section re-written

Line 777: Rephrased

Lines 814-819: Section rewritten

Lines 927-927: Section rephrased.

We hope that the Editorial Board finds our revisions adequate and satisfactory. If additional revisions are found to be necessary we are of course willing to consider such.

Oslo March 27th 2023

Roy H. Gabrielsen
emeritus professor

---

## Referee Report (RR1)

**Review (2ⁿᵈ round) of "Analogue experiments on releasing and restraining bends and their application to the study of the Barents Shear Margin"**

**Comments on the revised manuscript:**

Firstly, I congratulate the authors on a successful modeling study. The experiments performed were rigorous, and the results are highly insightful and information dense. I think that this is an important contribution to our understanding of the Barents Shear Margin and complex shear margins in general. Though the authors revisions mostly satisfied my initial comments, I suggest additional revisions be made:

**Main comments:**

Aside from lingering grammatical flaws (see comments below), the description of the experiments including methodology and results is satisfactory after the initial round of revisions. However, I suggest that the discussion is slightly revised both in terms of structure and content.

As it is, the discussion seems to address segment 1 in the experiment and its relationship with the Barents Shear Margin, then moves into a larger discussion around segment 2. Then there is the final paragraph of the discussion, which is overall unclear. Logically, segment 3 would be discussed here, yet there is potentially a typo that makes the entire paragraph confusing. Though not highly essential, I recommend the use of subheadings to improve the clarity of the discussion.

In terms of the content of the discussion, I find it to be largely descriptive and could be improved by discussing the more general implications of the observations within the scope of the questions addressed (timing of structural development, types of structures observed, sedimentary basin evolution). This could be done with a separate discussion (perhaps just a final paragraph) on the more general process of multiphase deformation in a complex shear margin. What are the major takeaways from the experiments that might be more ubiquitous, globally? How does this fit in the current literature of mixed-mode, multiphase deformation, and its structural and bathymetric expression? How is strain redistributed in the next phase of deformation - what is the role of pre-existing structures, lithological complexity, etc.? In some instances, the authors do this, for example discussing the role of mechanical stratigraphy on fold configurations (line 880). However, these experiments are extremely information dense, and more insights can be extracted from the results. Without something to this effect, I feel the impact of the article is limited to the scope of the Barents Margin.

Lastly, the conclusion could be improved by mentioning the outlook of the experiments. Some examples may be: what remains unresolved? what can be done next to understand margin and process? what techniques could be used to address these things?

**Grammar:** Though the authors mentioned they addressed this issue, the grammar throughout the manuscript is still very unpolished and, in many cases, detracts from the reader ability to understand what is written. Some examples (of many) are listed below:

- Lines: 1, 181-183; 236 – 237; 730 – 732; 739 – 741, 759, 578, 658 – 659, 838-842, 868, 1004, and several others).

**Figures:** The figures have been greatly improved since the initial submission; however I suggest some additional changes:

- **Figures 5 and 6:** segment labels be included on the side of panel A.
- **Figure 7:** a simple schematic could be added to show the location of segments as in Figure 4. Though Figure 4 contains all the required information, these changes (with additions to Figs. 5&6 mentioned above) would save the reader from continuously going back and forth throughout the paper.
- **Figure 9:** Though the meaning is somewhat obvious, the standalone letter "P" should be defined in the figure caption.
- **Figure caption 11:** The word "note" starts three sentences. For flow, it is a good idea to vary the sentence structure. Each panel (A-I) should be defined in the figure caption or combined into a single element.
- **Figure 12:** Panels A, B, and C (and elements of each panel) are not explained. Only panel A is referenced in-text.

**Textual comments:**

Lines 838 – 842: The meaning of this sentence is unclear. Do the authors mean that: In the margin underlain by continental crust, it is unlikely that enough opening occurred to link the basins?

Line 856: This the only location where Figure 12A is explained. Figure 12B,C is never referenced separately in the text.

Lines 925 – 926: How did sedimentary communication occur without basin linkage? It may be helpful if some processes were presented to explain this.

Lines 999-1000: This sentence is unclear. Do the authors mean a hybrid between segments 2 & 3? Or that segment 3 is a hybrid between segments 1 & 2. If not, how can a segment be a hybrid between itself and another segment?

---

## Referee Report (RR2)

[referee-annotated manuscript omitted]

---

## Referee Report (RR3)

Review (3rd round) of "Analogue experiments on releasing and restraining bends and their application to the study of the Barents Shear Margin"

General comments:

After a third careful review of the article *Analogue experiments on releasing and restraining bends and their application to the study of the Barents Shear Margin* I recommend that the article is once again returned for minor revisions. While I am satisfied with the previous revisions in terms of general content and figures, the writing remains grammatically poor with spelling errors scattered throughout the article. I marked minor revisions because this in many places hinders understanding on the part of the reader. In some locations it is impossible to understand what the authors are trying to communicate. A potential solution is that the authors submit the article for corrections from a professional scientific editing service. Examples (in reference to the tracked changes version) include lines 101 – 109, 128 – 130, 136, 154 – 157, 186-189, 219-220, 245, 252-259, 295, 402-405, 441-445, 458-462, 481, 505-508, 518, 549, 555-557, 576, 609, 623-625, Figure 8 caption, 686-687, 707-708, 715-716, 741, 778, Figure 9 caption, Figure 11 caption, 795-796, 812, 836-839, 861, 871-877, 885, 915, 955, 960, 1057-1059, 1092, 1097, 1106-1108, 1119-1123, 1157, 1173. Note that these are only the errors that I caught and noted, there are certainly more that I missed. Interestingly, it seems that many of these grammatical errors were introduced in the most recent revision, though some have been lingering since the first edition.

I would like to reiterate from my previous revisions that I think these experiments are valuable to better understand the Barents Margin and complex shear margins in general. I think the journal's readership will find the results to be interesting and insightful once the article is properly revised for grammar, flow, and spelling.

Other comments (in reference to tracked changes version):

Figure 2: TWT and Twt – inconsistent capitalization.

Line 549: Delete "from the early works"

Line 576: En-echelon what?

Lines 1113-1123: This paragraph is repetitive. The utility of analog experiments to understand the Barents shear margin has been emphasized in many locations already throughout the manuscript.

---

## Editor Decision (ED1)

I have found a few minor issues that should be corrected during the further proofing process:

**General:**

- Please check that all units are correctly attached to their respective quantities. Sometimes a line break separates units from their quantity. This is also the case for the numbering of some elements, e.g. sometimes phase 1 is split by a line break. Consider using a protected space ("Shift + Space" in MS Word or a fiducial "," in LaTeX). Usually, this is also checked by Copernicus' copy editing.
- Also sometimes there are spaces between dashes and values and sometimes not. E.g.: "3-4 cm" vs. "3 - 4 cm".
- Sometimes you are using abbreviations for specific units, sometimes their full name and for some structures you introduce an abbreviation but it is never used later on. This should be consistent throughout the manuscript. I know this can be confusing to the reader but I am also not sure if it is better to abbreviate the structures or to always give the full name. Both have advantages and disadvantages.
- Check for language consistency (American or British English), e.g. "analyzing" (US) vs. "analysing" (UK) and "analog" (US) vs. "analogue" (UK). Maybe this is also done by Copernicus.
- Check that all references have a DOI. This will definitely come up during copyediting...

**Specific (line numbers refer to the manuscript pdf version 5):**

L28: "... extension, and eventually..."
L35: doubling of "significant" and "significance"
L66: doubling of "kind of", second one can be removed
L87: split sentence here: "...the North Atlantic Ocean. Its configuration..."
L102: maybe remove "(initial)", as it also appears in the next sentence
L114+: "...Red box shows the study area. B) Structural map of the Barents Sea shear margin. Note the segmentation of the continent-ocean transition..."
L126: "...Results from the experiments..."
L128: split sentence here: "...deformation. Additionally, they allow to identify..."
L137: Please check the spelling of each of the structural elements in the whole manuscript, sometimes each word is capitalized, sometimes only the names, e.g. "de Geer zone" vs. "De Geer Zone" or "Barents Shear Margin" vs. "Barents Sea shear margin"
L161: "c." -> either "approx." or "ca." (also check other appearances in the manuscript, e.g., L407)
L183+: I think here "continent-ocean transition" is sufficient, as it was previously defined like this and is named "COT" again in the next paragraph.
L193: "...area east of the ..."

L198: "... coincides with the southeaster ..."

L209: "... which is interpreted..."

L210: double use of "which" in one sentence. Consider rephrasing.

L211: Duplication within the sentence, remove one of these: "has been associated with shear" or "affiliated with the development of the shear margin"

L212+: Maybe the sentence could be split here: "...(Riis et al., 1986). Although it was a positive structural element from the middle Cretaceous to the Pliocene, it may also have been activated earlier."

L218: "...splitting off of slivers..."

L223: "...even during the earliest..."

L228: "In particular, the hanging wall..."

L237: "...dense in the northern ..."

L239: "...study area, while previously published correlations provided calibration and age of each mapped seismic horizon."

L244: Please check your links. www.npd.com leads to an American consumer survey and trend monitoring company which has nothing to do with geoscience at first glance.

L246: Maybe rephrase to: "The folds are usually located in the hanging walls of extensional faults, and the fold traces and structural grain of the thick-skinned major faults are generally parallel."

L258: Maybe rephrase to: "...dynamic, so the ultimate architecture of such systems contains structural elements that did not emerge simultaneously."

L330: "...sheet represent the oceanic..."

L333+: Maybe rephrase to: "To simulate the ocean-continent transition, a sand wedge was made with a constant slope angle determined by the thickness difference between the intact and the stretched crust, covering the width of the silicon putty layer (Figure 3B)." I think that is easier to follow than the current sentence.

L340+: Maybe format the "Segment 1", "Segment 2" and "Segment 3" sentences as a list:
- Segment 1: ...
- Segment 2: ...
- Segment 3: ...

L411: "...crust (20-30 km, Breivik et al., 1998)."

L412+: "The brittle crust, represented here as dry feldspar sand, deforms (...) putty shows ductile deformation and folding."

L415: I think "configuration" fits better than "geometry" here.

L418: Maybe rephrase to: "After the experiments were completed, they were covered with a thin layer of sand to stabilize the surface topography before the models were saturated with water and cross sections were cut across the velocity discontinuity in a fan shape (Figure 3C)."

L430+: This might have been mentioned before and could be an unnecessary repetition.

L444: "...in the text (Figure 4)."

L446: "...(BarMar6) compared to ..."

L455: "...development are displayed..."

L457: "...morphology, as seen on the surface, were detected in the different stages of the

experiments."

Table 1: PSE-4/Orientation -> "Parallel master fault in restraining bend"

L478: Replace "very initial" with "first"

L479+: Rephrase: "In particular, in segments 1 and 3, there is a series of oblique en echelon folds between Riedel shear structures (PSE-1 structures) oriented approximately 135° (NW-SE) to the regional VD, which rotate in the NNW-SSE direction by continued shear."

L492+: Rephrase: "After a horizontal displacement of 0.25 cm in segment 1, which included releasing and restraining bends separated by a central strand of neutral shear, a slightly curved surface trace of a NE-SW striking, top-to-NW trending normal fault developed in the southernmost part."

L517+: "...that steepened downward, with the deepest parts..."

L521: "..shallowest sequences developed at a later..."

L527: no hyphens between "horse tail like", also true for some other combined words. I am not a native speaker, so it is also not always clear to me when to use a hyphen and when not.

L530: Rephrase: "The structuring in Segment 2 was determined by the precut crescent basement fault (velocity discontinuity), which caused the development of a releasing bend along its southern boundary and a restraining bend along its northern boundary (Figure 11)."

L582: Split sentence here: "...(see below). But these..."

L589: Move "however" to the start of the sentence: "However, the northern..."

L591: Mind the hyphens: "Thus, faults 6 and 7 delineated a growing, crescent-shaped, 5-7 cm wide thrust structure..."

L593: Rephrase: "Continued shearing gave these structures the character of an antiformal stack."

L602+: You could shorten "Riedel-" and "Riedel'-" to "R-" and "R'-", like you did previously.

L673: You could remove the parentheses: "In our experiments, the extension phase was followed by an oblique contraction parallel to the extension direction, as used in each experiment."

L674+: Split sentence: "More common, however, was the reactivation and rotation of faults that had developed during the strike-slip and extension phases. So was the development of isolated folds, often associated with inverted fault traces, producing structures such as snakeheads or harpoon structures (References et al...)."

L680: Rephrase: "The predominant structures associated with the contractional stage were still..."

L682: "...direction and subparallel to the pre-existing main fault systems ..."

L687: Here the colon could be replaced in "...length to amplitude ratio..."

L692: I think "attitude" should be replaced with "orientation"

L722: I think "jogs" should be replaced with "deflections"

L730: spacing between "...marked with "3"... "

L735: Rephrase: "Fault 3 continued to breach the basin fill even after the basin fill exceeded the original basin margin."

L738: Rephrase: "Note that figures "H" and "I" (bottom right) are viewed from different directions than the other figures."

L752+: Rephrase "Note that the fold axes are mainly parallel to the basin margins, but deviate in some cases in the central parts of the basins."

L783+: Split sentence here: "... Vestbakken Volcanic Province. The part formed by basin widening through sequential footwall collapse formed a shallower subplatform (sensu Gabrielsen, 1986) (Figure 11)."

L786+: Maybe rephrase to: "The master fault strand is accompanied by faults with significant normal throw on its hanging wall and is part of the larger Knølegga fault complex." Do I understand this sentence correctly?

L790: "2000 ms" I think milliseconds is wrong here, and why the 3000 meters in parentheses?

L797+: Maybe rephrase to: "They are located in both the footwall and hanging wall of the boundary faults and strike parallel to them, and the axes of these structures are parallel to the main faults."

L840+: Split sentence here: "...Knølegga fault complex. However, new transverse faults within the basin that had developed during the shear stage (see above) were also reactivated and contributed..."

L842: Rephrase: "It is unlikely that a stage..."

L844: Rephrase: "... may have occurred."

L882-899: I think the font has changed here, but Copernicus should correct that.

L882: Add comma between "...steeply inclide, closed to open..."

L884: No comma before "and". (Check also other occurrences in the manuscript)

L888: "...wavelengths on the order..."

L913: I think "attitude" should be replaced with "orientation"

L920: Rephrase "...these structures cannot be dated by seismic stratigraphic methods (onlap configurations, etc.), we assume that these fold extensions can be correlated with the tight folds..."

L926: Comma is separated from its word.

L929: Rephrase: "Because of the internal configurations, the three segments..."

L941: Rephrase: "By constraining the experimental model to the outline of the boundary geometry and introducing a dynamic stress model consistent with the latest understanding of regional tectono-sedimentologic evolution, we were able to interpret the observations from the reflection seismic data in a new light."

L958: Rephrase: "...,our model shows that the initial architecture of the edge is indeed important and..."

L984: Rephrase: "Comparison of seismic mapping and analogue experiments shows that one of the major challenges in analysing structural patterns in shear margins with complex geometry and multiple reactivations is the low potential for preservation of structures formed in the earliest stages of development."

---

## Author Response (AR2)

Dear Editors of Solid Earth,

We have submitted/posted a revised version of our paper: "Analogue experiments on releasing and restraining bends and their application to the study of the Barents Shear Margin» and hope the revisions make the paper ready for publication in Solid Earth.

We are indebted to the anonymous reviewer on the corrected version of our manuscript for his/her comments/suggestions concerning the scientific content and linguistic details as well.

**Abstract**

All simplifications as suggested by Reviewer have been acknowledged.

**Introduction**

The text has been improved on all points as suggested by the Reviewer.

Line 94: We disagree that the term "halokinesis" is uncommon. We use the term in accordance with the original definition by Trusheim (1957) and modern textbooks of salt tectonics (e.g. Jackson & Hudec 2017): Halokinesis: "Salt tectonics in which salt flow is powered by gravitational forces". The term is a standard term in textbooks and scientific literature on salt tectonics, and we have therefore chosen to keep it as it is.

**Description**

The text has been improved on all points as suggested by the Reviewer and all unclear statements have been rephrased.

The descriptions of all structural elements have been surveyed.

Particularly the description of the **Vestbakken Volcanic Province** has been revised. Unclear pints identified by the Reviewer have been straightened, and the architecture of the section has been adjusted.
Definitions have been added (eg. COT-margin).

The section on the **Senja Ridge** has been expanded with information requested by Reviewer.

Lines 397-418: The reviewer requests information on deformation velocities in the experiments: This information is given in lines 397-398 (constant $v=10$ cmhr$^{-1}$) after initial experiments demonstrated that changes in shear velocity did not considerably alter the geometry of the shear zone. An addition has been made better to explain Phase 3).

**Modelling results**

**Deformation phase 1**

*PSE-1-structures:* Figure caption Figure 8 expanded for explanation of location (segment 1) and fold geometry. Figure 8 was included after the pervious comments of the referee. The figure is extensively referred to in this section, and combined with an expanded figure caption, we now think that the characteristics of SPE-structures are now duly documented.

Some references to previous works are considered superfluous by the reviewer and have been taken out.

Lines 575-579: The description of the development has been rephrased and reference to Figure 4 has been added o enhance clarity.

Line 701-703: It has been explained above that experiment BarMar6 was used as a reference to evaluate the difference between orthogonal and the oblique angle of extension/contraction. We do not feel that is necessary to repeat this.

**Deformation phase 2**

Line 718-743: We apply the whole circle for describing the extension direction of the western plate. The eastern plate was kept in a fixed position during the experiments as explained in the Experimental set-up. Accordingly, extension angle of $315^0$ means NW-directed extension.
Description of widening of the strike-slip basins has been rephrased.
All items pointed to by the reviewer have been simplified and partly re-written t avoid unclear phrasings.

**Deformation phase 3**

Technical information on last phase of the experiments have been transferred to the set-up-chapter.
The section has been re-phrased on the points suggested by reviewer. Some statements regarded to be superfluous by the reviewer have been omitted.

Some figure references have been adjusted and/or added as requested.

**Discussion**

We agree with the reviewer that the experiments generated much new information on multistage continental margins in general and that it would be tempting and beneficial to include a full discussion on this. In the present discussion (as also pointed to by the reviewer) we have tried to explain he regional observations in light of the experiments. Because the manuscript is already voluminous, we however found that expanding the

discussion to a general evaluation of shear margins would explode the framework of the paper. We still acknowledge the reviewer's views on this and will consider writing a separate paper in the near future to cover these general aspects.

We have therefore tried to fulfill the reviewer's suggestions by:
1) breaking the discussion down into sections separated by sub-headings
2) restructuring and rephrasing parts of the discussion
3) adding a short final section on general aspects of the experiments/structuring of shear margins

**Figures/figure captions**

**Figures 5,6 and 7** have been expanded to ease reference between figures and the main text.

**Captions of figures 9,11 and 12** have been expanded and/or adjusted as suggested by the reviewer.

**Figure 13:** The reviewer requests a reference for this, but this is an original figure not previously published. The figure still is an expansion/up-date of a figure presented by one of the present authors (Faleide et al. 2008) and we have included a reference to his previous work).

We look forward to hear the final decision of the editors.

With best regards,

Roy H. Gabrielsen
(corresponding author)

---

## Author Response (AR3)

Dear Editor Solid Earth,

We do acknowledge the detailed reading and comments of the reviewer. We have tried to meet the reviewers concerns in the present update. Unfortunately, the time-frame with dead-line of 18 July does not allow for a full linguistic revision by a professional reader as suggested by the reviewer. We have therefore revised the manuscript ourselves to our best ability. Our revised manuscript (3rd revision) has been uploaded on the Solid Earth Net page. **Please note that the line numbering in the letter from the reviewer refers to the tracked changes version used by him/her and will therefore not match line numbering in the corrected manuscript.**

We have scrutinized the full and all specific comments from the reviewer have been acknowledged. We have particularly focused on shortening and simplifying of long and complex phrasings and sections. Thus, we have made some grammatical corrections/simplifications in addition to those identified by the reviewer and corrected some minor printing errors. Our revision includes the following: We have

Identified and corrected some minor typos
Deleted some words that were duplicated or in excess
Added a few words that were missing
Split and shortened several very long sentences/sections
Adjusted style of headings to make sure that they become systematic and in accordance the structure/hierarchy of the paper (1st-order, 2nd-order, ..)

Please note that we have consequently used the abbreviation "e.g." (exempli gratia) throughout the manuscript, because this format (rather than "eg.") is the standard recommended in Webster's dictionary.

Lines 6-7: The name of one of the authors have been misspelled in the heading of the manus (corrected in this version): Gi**e**nnenas should be Gi**a**nnenas. References and reference list have been corrected accordingly.

With reference to the points of improvements specified by the reviewer, the following corrections have been performed:

Lines 101-109: Long sentence has been simplified and divided into shorted sentences.

Lines 128-130: Sentence rephrased.

Lines 154-157: Introduction to section is rephrased.

Lines 186-189: Misprint has been corrected.

Lines 219-220: Sentence has been simplified.

Line 245: Misprint corrected.

Lines 252-259: Rephrased

Lines 441-405: Reflection seismic data-chapter (p.8): Unclear statements are rephrased.

Experimental set-up (lines 402-549): Adjustments of unclear points in text.

Line 518-557: Phrasings simplified and specified.

Line 576: Sentence adjusted

Line 609: Superfluous reference removed

Lines 623-625:  We find that this description is correct and in accordance with descriptions in the literature (Woodcock & Fischer 1986; Woodcock & Schubert  1984 and Twiss & Moores 2007) and have kept the phrasing as it is.

Figure 8: Caption rewritten

Lines 686-687: This phrase describes well the observations of the experiments, so we have kept it as it is.

Figure 9: Caption rewritten

Figure 11: Caption adjusted. Misprint concerning color of N-arrow corrected

Lines 795-1173: Several sections have been simplified/re-written to avoid unclear phrasings and explanations.

Lines 1113-1123: Albeit repetitive, we have kept the introduction to the Discussion as it is, because we think this is relevant for the reader who jumps directly to the conclusion.

Figure 2: Inconsistency regarding terms TWT/Twt has been corrected

We hope that these corrections are acceptable by the Editorial Board and look forward to see the final version. Thank you very much for your patience and cooperation.

Best regards
Roy H. Gabrielsen
On the behalf of the authors

---

## Author Response (AR4)

Dear Editor Solid Earth,

Again, we have to thank the reviewer of our paper **Analogue experiments on releasing and restraining bends and their application to the study of the Barents Shear Margin** for a very detailed and concerned review. We are certain that most suggestions have enhanced the readability of the paper. We have therefore generally acknowledged all corrections/rephrasings suggested (splitting of sentences etc).
We have also changed some US spellings with UK-standards.

For the items where we have chosen not to follow the reviewer's recommendations in full , we have the following comments:

1) We have kept phrasings and terms, which have established (formal) defintions, or are in common use in structural geology literature.

2) Acknowledging the recommendations of the International and Norwegian Stratigraphical Committees, we separate between formal and informal names for stratigraphic units and structures by using recommended codes of spelling, so that formal terms are capitalized (Barents Shear Margin) and informal names are not (Hornsund fault system).  We have corrected terms, which have been incorrectly or inconsequently written by mistake (e.g. Barents shear margin).

3) We have the understanding that terms "approximately" and "circa" (abbreviation "c.") are of equal meaning and use in English, and have kept both terms where we find that each terms fits the actual phrasing the best.

4) We do not find that terms boundary and transition are equal, since the former is used in a regional (large-scale, unspecified) context, whereas the latter characterize the nature of the structure: When we use both (e,g, "continent-ocean boundary/transition"), we mean a regional boundary with a transitional geometry.  COT and COB are term for a boundary with a transitional geometry commonly used in the literature.

5) In the cases where we feel that the rephrasing suggested by the reviewer will change the meaning of the sentence and have kept the phrasing as it is as it is.

6)The reviewer suggest that the presentation of the segments (lines 340ff) is given in the format of "a list" (without re-writing) rather that included as a continuous section. We feel this would add an unnecessary complex layout to the section, and that this is a decision that should be made by the Technical Editor: We have therefore not changed this.

7)We have not changed the terms Riedel- and Riedel'-structures to R- and R'-structures as suggested by the reviewer, since we find this decision should be taken by the Technical Editor and harmonized with the standards of Solid Earth,

WE do hope that the manuscript is now in full order for printing in Solid Earth,

Sincerely,

Roy H. Gabrielsen
Professor emeritus